# From Nanobiotechnology, Positively Charged Biomimetic Dendrimers as Novel Antibacterial Agents: A Review

**DOI:** 10.3390/nano10102022

**Published:** 2020-10-14

**Authors:** Silvana Alfei, Anna Maria Schito

**Affiliations:** 1Department of Pharmacy (DiFAR), University of Genoa, Viale Cembrano 4, I-16148 Genova, Italy; 2Department of Surgical Sciences and Integrated Diagnostics (DISC), University of Genoa, Viale Benedetto XV, 6, I-16132 Genova, Italy; amschito@unige.it

**Keywords:** antibiotic resistance, novel antimicrobial agents, cationic antimicrobial polymers, non-specific membrane disruption, biomimetic cationic dendrimer nanoparticles, amino acid-modified polyester-based dendrimers

## Abstract

The alarming increase in antimicrobial resistance, based on the built-in abilities of bacteria to nullify the activity of current antibiotics, leaves a growing number of bacterial infections untreatable. An appealing approach, advanced in recent decades, concerns the development of novel agents able to interact with the external layers of bacteria, causing irreparable damage. Regarding this, some natural cationic antimicrobial peptides (CAMPs) have been reconsidered, and synthetic cationic polymers, mimicking CAMPs and able to kill bacteria by non-specific detrimental interaction with the negative bacterial membranes, have been proposed as promising solutions. Lately, also dendrimers were considered suitable macromolecules for the preparation of more advanced cationic biomimetic nanoparticles, able to harmonize the typical properties of dendrimers, including nanosize, mono-dispersion, long-term stability, high functionality, and the non-specific mechanism of action of CAMPs. Although cationic dendrimers are extensively applied in nanomedicine for drug or gene delivery, their application as antimicrobial agents is still in its infancy. The state of the art of their potential applications in this important field has therefore been reviewed here, with particular attention to the innovative case studies in the literature including also amino acid-modified polyester-based dendrimers, practically unexplored as membrane-active antimicrobials and able to kill bacteria on contact.

## 1. Introduction

The increasing growth of resistant bacterial strains, which represent a highly worrying trend that has characterized the last few years, has caused the appearance and the re-emergence of serious infections, in particular in nosocomial settings [1]. In this regard, pneumonia, bloodstream infections, wound or surgical site infections, and meningitis are often associated with the failure of antibiotic-based treatments or with the concomitant lack of new antimicrobial agents, [2,3,4]. Gram-negative bacteria such as *Klebsiella pneumoniae*, *Acinetobacter baumannii*, *Stenotrophomonas maltophilia*, *Pseudomonas aeruginosa*, *Burkholderia cepacia, and Escherichia coli* pose a major threat to human health, since they are the most critically resistant and rapidly spreading bacteria [2,5,6,7]. Moreover, in addition to their intrinsic resistance mechanisms, these pathogens are rapidly becoming multidrug-or even pan-drug-resistant to most life-saving drugs [1]. In particular, aerobic non-fermenting Gram-negative bacilli such as *A. baumannii*, *P. aeruginosa*, and *S. maltophilia*, are emerging as clinically relevant superbugs, contributing significantly, with their alarming resistance levels, to numerous therapeutic failures [1]. Gram-negative bacteria, unlike Gram-positive bacteria, are characterized by high and similar resistance levels, both in Europe and in the United States [2,7].

Given this situation, as well as being inspired by reports developed by a group of independent experts led by World Health Organization (WHO) [5,6,7], the medical research community must develop new antimicrobial agents active on current resistant strains of Gram-positive and Gram-negative bacteria [2]. Furthermore, because bacterial infections, especially if caused by biofilm production, hinder the durability, reliability, and performance of many medical devices and implants, antibiofilm strategies such as antibacterial coatings that repel bacteria and prevent biofilm formation are highly desirable [3].

Natural cationic antimicrobial peptides (CAMPs) are a class of unconventional antimicrobial agents with a broad spectrum of action, active on a wide variety of Gram-positive and Gram-negative bacteria, fungi, protozoa and yeast [2]. These molecules, without the need to enter the bacterium cell or to interfere with specific metabolic processes, basically act based on their positive charge. Their action is rapid and not specific and is based initially on electrostatic interactions with the bacterial surface, followed by the progressive damage of the bacterial outer and/or cytoplasmic membranes (OM and CM) that leads to the bacterial death. On the base of this mechanism of action, CAMPs kill pathogens simply through external contact, without the need to address the numerous resistance mechanisms due to genetic mutations that bacteria can develop [2]. In other words, since these materials do not interfere with the vital processes for bacteria, which can eventually be modified—by genetic mutation—by resistant pathogens, they are generally indifferent to the multiple resistance mechanisms developed by the bacteria on which they act. Recently, cationic antimicrobial macromolecules, inspired by natural CAMPs, have gained increasing attention from the scientific community because, compared to small molecules of drug, they possess several advantages such as higher long-term activity, limited residual toxicity, chemical stability, non-volatility and the inability to permeate through the skin due to their macromolecular structure and high molecular weight (MW) [2,8]. Among polymers, dendrimers (Ds), a specific class of nanoscaled, hyperbranched, tree-like macromolecules, with a symmetric well-defined structure and a three-dimensional architecture [9,10,11,12], have recently shown to function as antibacterial agents and as antimicrobial surface coatings as well [3]. The first synthesis of dendrimer materials, whose structure was conceptualized in the early 1970s, dates back to the mid-eighties [3]. Although Ds, including positively charged ones, such as cationic poly(amidoamine) (PAMAM), polypropylenimine (PPI), dendritic polylysine, and peptide structures, have been actively investigated for a wide range of industrial and biomedical applications, their potential usages as antimicrobial agents mimicking CAMPs, both as drugs, as surface coating agents and drug-delivery systems, has been recognized only very recently [3].

In this regard, based on the Scopus data, the scientific interest in cationic polymers as novel antimicrobials was limited until 2000, but has grown steadily and exponentially to date [2]. On the contrary, the trend of scientific production and research in the field of antimicrobial dendrimers (ADs) over a period of 30 years definitively underscores how this was non-existent until 2007, then it started to increase, but not constantly, with the highest production in the last decade (Figure 1).

Furthermore, as shown in Figure 2, a more restricted investigation of the scientific output of the last decade concerning cationic antibacterial dendrimers (CADs) (yellow bars), the subject of this work, revealed that the actual number of studies is far lower than that reported in Figure 1 (purple bars), which generally depicts the number of publications on ADs.

However, different types of Ds, including cationic ones, have been developed [3,13] for the treatment of infections sustained by multidrug-resistant bacteria, mainly during the last decade. In this context, some synthetized antimicrobial Ds also proved to have antibiofilm effects [14] or capabilities to act in synergism with commonly used antibiotics [15,16], and in some case studies in vivo evaluations were also performed [17].

Commercially available PAMAM and PPI dendrimers (PAMAM Ds and PPI Ds) are the most investigated, but despite their considerable broad-spectrum activity observed in vitro, if not opportunely modified to tone down several issues, such as low biodegradability, susceptibility to opsonization, toxicity to mammalian cells, including hemolytic toxicity, cytotoxicity, and hematological toxicity, as well as fast clearance [9,10,11,12], in their native form they are not suitable for clinical applications. To address these issues, as in other biomedical applications, uncharged dendrimer matrices, decorated with protonable residues [1,18], are worthy of consideration as a less toxic alternative [19]. In this context, amino acid-modified, polyester-based dendrimer scaffolds should be the most attractive, also because of their good biodegradability [1,18].

In this paper, after a short overview of the different classes of antimicrobial Ds developed up to 2010, the state of the art of cationic ones developed in the last decade, has been reviewed, highlighting the relationships structure/activity with particular attention to how dendrimer surface groups may affect the antimicrobial activity and their capability to inhibit the biofilm formation. Case studies for each dendrimer class have been also included and summarized in Tables. Interestingly, in this analysis the few examples of amino acid-modified polyester-based Ds, so far practically unexplored as membrane-active antimicrobials and able to kill bacteria on contact, have been included in a review for the first time [1,18].

## 2. Biofilm as a Mechanism for Antibiotic Resistance

The ability of a bacterium to resist the damaging activity of an antimicrobial agent can be inferred by intrinsic mechanisms, based on the innate ability of a bacterial species to elude the activity of a particular drug, through its inherent structural or functional characteristics. More frequently, nevertheless, insusceptibility to a drug (or several drugs) is genetically determined, and can be transmitted across the same pathogen or even to similar bacterial species, by passing the DNA responsible for the mechanism of resistance through several complex mechanisms such as conjugation, transformation and even transduction [20].

An alternative mechanism capable of conferring antibiotic resistance can be provided by the ability to produce biofilm, a multi-shaped strategy of bacteria to protect themselves from host defense, disinfectants, and antibiotics. Biofilm formation enables single-cell organisms, planktonic microorganisms, to embark on a temporary multicellular lifestyle, sessile forms, in which “group behavior” facilitates survival in adverse environments [21]. The expression of surface molecules, nutrient use and virulence factors are altered by this reprogramming and bacteria become endowed with an extra arsenal of properties that enable their survival in unfavorable environments [22]. Within the biofilm, sessile bacteria are cocooned in a self-produced extracellular matrix, which accounts for 90% of the biofilm biomass.

The biofilm biomass is made of extracellular polymeric substances (EPSs) including a complex assemblage of enzyme proteins, polysaccharides, such as cellulose, polyglucosamine (PGA) and exopolysaccharides, genetic material, as extracellular DNA (eDNA), anionic and cationic glycoproteins and glycolipids, which enable intercellular interactions, keep bacteria in close proximity with each other, serve as a connecting agents, and stabilize the scaffold for the three-dimensional biofilm structure [20,21]. In the matrix, nutrients are trapped for metabolic uses by the resident bacteria and enzymes secreted by the bacteria modify EPS composition in response to changes in nutrient availability, while water is efficiently retained through H-bond interactions with hydrophilic polysaccharides [21].

Bacterial aggregation and subsequent biofilm maturation is determined by reversible and irreversible stages and involves numerous factors. The first step consists of the introduction of bacteria to a surface, followed by adherence mediated by extracellular adhesive appendages and secreted adhesins [21]. Adherence of bacteria is progressive and leads to an exponential growth of the number of colonizing bacteria, to changes in gene expression and to up-regulation of factors favoring sessility, such as those implicated in the formation of EPS. Improved in number, sessile bacteria produce a high number of auto-inducers (AIs), i.e., signal molecules that are known as quorum sensing (QS) systems. QS mechanisms exist in Gram-positive and Gram-negative bacteria, but the signal molecules used to transmit information in the two groups are different. However, QS allows bacteria to synchronize and act as a unique great workforce that actively exchanges and shares products that play a pivotal role in maintaining biofilm architecture and providing a favorable living environment for the resident bacteria [20,23].

Bacteria can easily grow undisturbed in biofilms on a wide variety of surfaces and attach to inert or alive surfaces, including tissues, industrial surfaces, and artificial devices, such as catheters, intrauterine contraceptive devices, and prosthetic medical devices, implants, cardiac valves, dental materials, and contact lenses [20,23]. Due to the protective action of biofilm, bacteria that growth in the biofilm conditions (sessile form), are much more resistant to antimicrobial agents than bacteria that growth in normal conditions (planktonic forms) and their susceptibility to antibiotics dramatically decreases. In addition, their capability to survive in conditions of hostile environments, such as osmotic stress and metal toxicity, is highly augmented in the biofilm.

Moreover, biofilm-specific characteristics limit the diffusion of drugs, making it difficult to reach the bacteria, and nullify antibiotic activity [24]. To counteract biofilm-forming microorganisms, an antimicrobial agent must overcome several additional obstacles, such as an increased number of resistant mutants, high cell density, molecular exchanges, substance delivery, efflux pump, persistent and dormant cells, altered bacteria growth rate and different gene expression [20,25]. In this regard, Table 1 reports the mechanisms through which biofilms hamper the activity of antibiotics with examples of microorganisms producing biofilms and the related inactivated antibiotics.

The presence of biofilm renders antibiotic treatments ineffective at the doses that would instead be effective on the bacterium in its planktonic form. The formation of biofilm represents an important virulence factor, which by allowing the colonization of living tissues or medical devices and protecting the bacteria that constitute it from the action of chemotherapy, plays an important role in the pathogenesis of many bacterial, subacute and chronic diseases, typically recalcitrant to antibiotic therapies. Furthermore, since many infections are of polymicrobial origin, biofilms can also be made up of different bacterial species. In this context, despite mechanism being unknown, polymicrobial biofilms are associated with an even higher level of antimicrobial resistance and, therefore, with serious episodes of therapeutic failure. In this regard, it was determined that in vivo *P. aeruginosa* growing in a mono-species biofilm is twice more sensitive to gentamicin antibiotic than that growing in a multispecies biofilm encompassing *S. aureus*, *Enterococcus faecalis*, and *Finegoldia magna* [40]. Otitis media is a biofilm-mediated multi-microbial infection sustained by *Moraxella catarrhalis* and *Streptococcus pneumoniae* commonly treated by amoxicillin-associated β-lactamase inhibitors or azithromycin. Nevertheless, it was determined that in the biofilm consisting of two bacteria species, *M. catarrhalis* produces a β-lactamase that renders *S. pneumoniae* resistant to amoxicillin, while reciprocally, *S. pneumoniae* protects *M. catarrhalis* from azithromycin by means of an unknown mechanism [41].

In a study concerning *P. aeruginosa* resistance to very active natural CAMPs, as well as to polymyxin B and colistin, it was proved that *P. aeruginosa* cultured in a biofilm with *S. maltophilia* have reduced vulnerability compared to *P. aeruginosa* single-species biofilms [42].

In a mature biofilm, sessile bacteria work together actively to maintain and care for the protective architecture of the structure in which they are embedded, but at appropriate times, the dispersion of the community may become an appealing option. Bacteria of a biofilm, in fact, have developed the ability to perceive environmental changes and evaluate whether it is still beneficial to reside in the biofilm or if it is time to resume a more convenient planktonic lifestyle. Biofilm dispersal can be the result of several causes, such as alterations in nutrient availability, oxygen fluctuations and increase of toxic products, or other stress-inducing conditions. Among the various signal molecules implicated in the shift between sexility and motility in bacteria of a biofilm, the universal 3,5-cyclic diguanilic acid (c-di-GMP) and a/the c-di-GMP binding protein, BdcA are widely used by species such as *P. aeruginosa* and *E. coli* [21]. Moreover, specifically in the case of *P. aeruginosa* and *E. coli*, additional dispersal mechanisms have been elucidated. These include the enzyme alginate lyase, the carbon storage regulator A (CsrA) protein, the production of surfactant molecules, the increase in rhamnolipid levels and, finally, the strategy to build microcolonies with a central void/vacuum zone, created by the cell death of the bacteria that make up the biofilm. A sophisticated dispersal mechanism has been observed for *B. subtilis* that forms robust biofilms, which lose their integrity after 5–8 days. The biofilm disassembly is facilitated by a mixture of *D*-amino acids produced during the stationary phase of growth and get incorporated into the peptide side chains of peptidoglycan in place of the terminal *D*-alanine, thus interfering with the anchoring of adhesive fibers on the cell surface, leading to fiber dissociation and loss of bacterial adherence [21].

Based on this scenario, exploring alternative cure options for biofilm-associated infections is an urgent mission. Although few innovative and effective antibiotic strategies have been designed, such as innovative techniques for dispersing biofilms, associations of conventional antibiotics with QS inhibitors and a mixture of these novel strategies, this research area is practically unexplored and far from undergoing clinical research and entering the commercial market. In this context, some antimicrobial Ds have proven antibiofilm effects, the capability to penetrate EPS [14], as well as to prevent biofilm formation, and might represent novel promising substances to counteract this alarming form of bacterial resistance [14,43,44].

## 3. Dendrimers (Ds)

Starting from 1978 [45] a series of differently structured dendrimer scaffolds, endowed with different physicochemical features, have been prepared and applied in several industrial and medical areas. The main types of dendrimer structures developed in the past 40 years are shown in Table 2.

From a structural point of view and in regard to the topics of this paper, some of the represented Ds already possess primary, secondary, tertiary, or quaternary nitrogen atoms and intrinsic cationic character without the need for extra functionalization. By contrast, the uncharged dendrimer scaffolds, need further conjugation with positively charged moieties, to be transformed into cationic macromolecules. Generally, cationic Ds, long before being considered to be membrane-active antimicrobial agents, have always attracted interest for biomedical applications because they possess the ability to penetrate cell membranes by endocytosis [46] and to escape the inactivating endosomal/lysosomal pathways via the proton-sponge effect [47]. In this regard, the proton-sponge outcome is allowed by an adequate buffer capacity of Ds, crucial property to survive the lysosomal attack that would translate into the premature destruction of Ds.

### Dendrimers as Antimicrobials

In the past, available antibiotics allowed to counteract severe infections successfully, thus reducing hospitalization and mortality [3]. Unfortunately, from the 1980s onwards, the number of infectious diseases associated with death started to increase and presently these conditions are responsible for the greatest number of deaths in developing nations. The increase in mortality rates, related to the emergency of non-curable bacterial infectious diseases, underlines the incessant development of resistance strategies to drugs activated by bacteria through the transmission or the acquisition of new genetic material, not only between strains of the same species, but even between strains of different species. To reach this goal, various classes of Ds that can inhibit microbial pathogens (even by killing them), have been designed. Their development, nevertheless, is still limited (Figure 1 and Figure 2) and their application in therapy is still in the exploratory phase. Interestingly, such dendrimer devices can act either as antimicrobial agents [3,13,17], drug-delivery devices [15,16,43], or bacteriophobic coatings [43,44].

#### Principal Types of Antimicrobial Dendrimers Developed in the Previous Decade

Although the aim and main purposes of this document is to review the state of the art of cationic antimicrobial Ds, in order to have a more complete view of all the types of Ds that possess the suitable structural characteristics to function as antimicrobial and/or antibiofilm compounds, this section quickly introduces those developed in the previous decade and already reviewed by Castonguay et al. (2012) [13] and by Mintzer et al. (2012) [3]. These include glyco Ds, cationic Ds, anionic Ds, and peptide Ds, which in some cases can also be either surface-adsorbed or metal-conjugated. Interestingly, cationic PAMAM Ds, before being considered to be per se active antimicrobial Ds, were mainly investigated as drug carriers to solubilize and deliver conventional antibiotics with synergistic intents [15,16,48].

In their relevant paper, Mintzer et al. [3] focused on the mechanisms of action of the different types of Ds, concentrating in particular on the effects of multivalence deriving from the tree-like and generational structure of the Ds, and also on the influence of the abundance of the active functions on the antibacterial potency of the developed dendrimers [3]. In this manuscript, the information reviewed by Mintzer and colleagues [3] has been summarized in a single Table (Table 3) to allow for a more rapid comparison of the essential characteristics of the various dendrimer nanoparticles endowed with antimicrobial activity.

## 4. Cationic Antibacterial Dendrimers (CADs)

After the promising results of cationic polymers as antimicrobial agents capable of mimicking the destructive and non-specific action of CAMPs [2], in recent years cationic dendrimer nanoparticles have emerged as promising new antibiotic agents [13,49].

The antibacterial activity of cationic material such as CAMPs and of their mimics depends on the electrostatic interaction between the positive charges of the device and the anionic bacteria cell surface, the progressive permeabilization of the bacterial membranes, the disruption of the lipid bilayer, the lack of cytoplasmic content, vital ions and cell death due to the disruption of the lipid bilayer [2,50]. Thus, similarly to other ADs, in CADs, the multivalence in terms of positive charges plays a key role in their antimicrobial activity, and high-generation cationic Ds proved to be biocides with high activity, capability of bring localized in specific organs reduced systemic toxicity and increased duration of action [50]. Research on these agents, in fact, focuses both on their intrinsic antimicrobial activity and on the possibility they offer as drug release systems, since they are suitable for encapsulating or covalently connecting biologically active agents [10,11,12]. This latter strategy improves the solubility of poorly soluble antibiotics, protects them from fast metabolism, increases their residence time in the circle, targets the drugs transported to specific sites of action and achieves synergistic cooperation between the cationic dendrimer carrier and the antibiotic, thus allowing a reduced dosage [2,15,48]. According to Scopus, although some sporadic case studies were reported before [3,51,52], the interest in cationic Ds started in 2005 and most of the research and scientific production belongs to the last decade.

Among cationic Ds, PAMAM Ds were broadly studied [15,48,50,53], PPI Ds were principally considered when the interest in cationic Ds started, while a limited number of case studies concerning PEI-Ds were published [54,55,56]. To achieve a fair compromise between activity, biodegradability, selectivity for bacterial cells and limited hemolytic toxicity and cytotoxicity towards eukaryotic cells, poly (lysine) [57] and peptide Ds [58] were extensively prepared and evaluated. Curiously, even though they were considered to be very attractive for biomedical applications as highly biodegradable and low cytotoxic [11], only very recently, polyester-based scaffolds, peripherally cationic for the presence of amino acids, have been taken into consideration as novel antimicrobial devices [1,18]. These Ds, by presenting an uncharged hydrolysable matrix, capable of better balancing the density of charge on their surface, maintain a strong antibacterial activity associated with reduced toxicity [1].

For each year of the last decade (*x* axis), Figure 3 shows the main classes of CADs that have been developed with the number of correlated studies (*y* axis), reflecting the researchers’ interest for the different dendrimer structures, over a 10 year period. It can be observed that although a high number of studies have been reported on cationic peptide Ds, dendrimers such as PAMAM Ds and peptide Ds have been developed throughout the decade. On the contrary, other structures, such as ammonium-terminated Ds (AT-Ds), including AT phosphorous and carbosilane Ds [59,60], attracted the interest of scientists only after 2015. Other Ds, such as PPI Ds, after initial consideration, were neglected, while PEI-Ds and organometallic Ds (OM-Ds), were taken into consideration only marginally. Polyester-based amino acid-modified Ds (PEAAM-Ds) have been considered and studied as antimicrobial devices only recently. However, all the cationic Ds reported in Figure 3 showed significant antimicrobial activity.

### 4.1. PAMAM and PPI Dendrimers (PAMAM Ds and PPI Ds)

Traditional and unmodified PAMAM Ds and PPI Ds possess a wide number of peripheral protonable primary amine groups and additional protonable tertiary amine groups in the inner matrix, which give them high antibacterial activity due to their high density of positive charge [15,48,50,53]. Surprisingly, even though PAMAM Ds were one of the earliest to be synthesized and extensively studied for various biomedical applications, their activity as cationic antimicrobial agents has only recently been evaluated.

Table 4 highlights how the number of functional groups of amines in the structure doubles, following the increase of one generation in the PAMAM Ds and PPI Ds.

Actually, the existence of a correlation between the generation number of PAMAM- or PPI Ds and their antibacterial activity is still under debate, and, according to some authors this correlation cannot be discerned, since both small and high-generation PAMAM Ds have shown considerable activity [3]. As for PPI Ds, an incomplete correlation between these characteristics was found by Cooper et al. who assessed both the influence of the generation number and that of the length of a hydrophobic chain, introduced into the structure of the molecule and considered responsible for the antimicrobial activity against *E. coli* [52]. The effect of the generation number on antimicrobial activity showed that G5 > G4 > G1 > G2 > G3. This effect was attributed to the balance between the higher potency achieved with a greater number of quaternary amines and the decreased permeability of the larger dendrimer analogs. The effect of the hydrophobic chain length on antimicrobial activity showed that C10 > C8 > C12 > C14 ≈ C16. The authors rationally attributed this trend to the dual binding site theory, which suggests that on the bacterial cell surface dual sites exist binding with antibacterial agents, for which the relative binding activities of long and short hydrophobic ligands differ [52]. In this context, six poly(quaternary ammonium) chloride Ds were synthesized by functionalizing a PPI D with alkyl chains of various lengths obtaining a series of cationic Ds whose structures encompass both the amphiphilic character of natural CAMP and the multivalence of Ds [61]. In vitro antimicrobial activity was assessed by measuring the inhibition zone diameter by agar-well-diffusion method, and the observed values have been reported in Table 5. Even though the authors indicated that the amphiphilic PPI-base Ds with shorter alkyl chains are potent antimicrobial agents with activity against Gram-positive and Gram-negative bacteria, including multidrug-resistant pathogens, such as the methicillin-resistant *S. aureus*, the assertion is questionable. As for reference data such as diameters of the inhibition zone of known antibiotics were not provided in the study, so that only a comparison between the synthetized Ds is possible. Furthermore, the concentrations of Ds used in that study (10 mg/mL) are extraordinarily high if compared to that of antibiotics (μg/mL). Finally, to assess the real applicability of these interesting Ds, investigations regarding their toxicity towards eukaryotic cells are lacking in the study, but mandatory.

Moreover, also results obtained from comparison studies between antimicrobial Ds and hyperbranched polymers reveal conflicting outcomes. Concerning PPI Ds studied by Cooper and colleagues [52], when colonies of *E. coli* were exposed to 12 μg/mL of a PPI D and to the same amount of a hyperbranched polymer, with the same number of functional groups and similar MW, all bacteria were destroyed in the first case while in the second case only a reduction of 30–40% was observed. By contrast, the hyperbranched carbosilane polymers, investigated in another study and reported by Mintzer in 2012 [3], showed higher activity than the second- and third-generation carbosilane Ds with similar molecular weight and ζ potential, being the MIC values of 8–16 μg/mL vs. 16–64 μg/mL against *E. coli* and of 4 μg/mL vs. 4–8 μg/mL against *S. aureus* [3].

Interestingly, in the earliest studies, PAMAM Ds were not evaluated as active antimicrobial agents *per se*, but rather as carriers for antimicrobial agents, with the purpose of improving their performance [3].

In this context, Neelgund and colleagues reported the design of antimicrobial nanohybrid Ds, namely f-MWCNTs-CdS and f-MWCNTs-Ag_2_S, synthetized by covalent grafting of cationic hyperbranched dendritic PAMAMs onto multiwalled carbon nanotubes (MWCNTs) followed by deposition of CdS and Ag_2_S quantum dots (QDs), respectively [62]. The antibacterial activity of f-MWCNTs-CdS and f-MWCNTs-Ag_2_S hybrid Ds (20 μg/mL) was evaluated on *E. coli*, *P. aeruginosa* and *S. aureus* and it was found that the biocidal action of MWCNTs, already improved by grafting of PAMAM, was further enhanced by loading CdS and Ag_2_S QDs. The percentages of bacteria growth reduction have been reported in Table 5, together with the hypnotized mechanisms of action based on the synergistic action of the PAMAM residues and the transported CdS and Ag_2_S QDs. In particular, the Ag_2_S Ds were more active than the CdS ones and both materials were more active against Gram-negative bacteria than towards the Gram-positive specie *S. aureus*. A little perplexity on the efficacy of f-MWCNTs-CdS and f-MWCNTs-Ag_2_S hybrid Ds prepared by Neelgund and colleagues arises from the fact that no reference data such as the growth inhibition activity of known antibiotics have been provided. In addition, to assess the real applicability of these interesting Ds, investigations regarding their toxicity towards eukaryotic cells are still lacking.

Despite their established antimicrobial activity, various characteristics limit their potential use, such as the long synthesis process, necessary for high-generation compounds and their non-specific interaction with the negative charges of bacterial or eukaryotic membranes responsible for cytotoxicity and hemolytic toxicity [11,53]. In addition, fast clearance from the blood circulation, as well as a high level of uptake in the reticuloendothelial system affect their activity [11]. To address these issues, different types of structural modifications or synthetic combinations of PAMAM Ds with more hydrophobic Ds have been developed. In this regard, PAMAM-based dendrimers compounds, with tunable cationic and hydrophobic characters able to self-assemble, were prepared combining poly(aryl ether)-based dendrons with PAMAM-based dendrimer residues, which are different for generation and terminal groups [53]. The obtained poly(aryl ether)-PAMAM-based amphiphilic Ds were assessed for their intrinsic antibacterial activity by determining MIC and MBC values and by time-killing experiments performed at 4× MIC concentrations on *S. aureus* and *E. coli* strains. The best performing Ds were, as expected, the amine terminated ones but interestingly, the best between the two was that of the lower generation. The MIC and MBC values of this dendrimer, namely AD-1, have been reported in Table 5. Time-killing experiments established that AD-1 was the dendrimer able to cause the high reduction in the growth of both bacteria along 24 h of exposure, thus confirming its strongest bactericidal effects. The cytotoxicity essay performed on AD-1 highlighted that NIH/3T3 fibroblast cells showed significant damage only when they were exposed to AD-1 concentration four times higher than MIC or to an AD-1 dosage 2-times the MBC for both *E. coli* and *S. aureus* species.

A highly cationic dendrimer, containing both a positively charged fourth-generation PAMAM architecture and a quaternary ammonium compound covalently linked, known as 1-hexadecyl-azoniabicylo [2.2.2] octane (C16-DABCO), and possessing antibacterial activity *per se*, was prepared [43]. Additional mannoside end groups have also been incorporated into the antimicrobial dendrimer, to prevent the bacteria adhesion to the surface of eukaryotic cells by an intercepting mechanism [43]. First, in the study of Vankoten and colleagues, the antibacterial activity of the obtained materials was tested against several Gram-positive and Gram-negative strains of pathogenic bacteria. With the exception of *Streptococcus oralis*, which showed MIC values > 20 μM, the remaining species were reported to be quite susceptible to the dendrimer, displaying MIC values in the range 0.13–2.0 μM. Unfortunately, the observed hemolytic toxicity (MHC = 0.6 μM) limits the clinical use of C16-DABCO dendrimer for the treatment of infections caused by *S. aureus* and *B. cereus* (MICs = 0.13 μM). A further study evaluating the tendency of the prepared dendrimer to select resistant strains within a specie was performed on *E. coli* and *B. cereus*. The results established that the multivalent antimicrobial dendrimer, due to its ability to deliver a detrimental dose of positive charge to the bacteria, displayed a significantly reduced tendency to develop bacterial resistance [43]. C16-DABCO dendrimer has also been tested for its ability to disrupt biofilm and/or inhibit biofilm formation. No inhibition or reduction of biofilm biomass was observed in any of the species tested [43]. Interestingly, experiments aimed at evaluating the prevention of biofilm formation, proved that on membranes pretreated with a 1 mg/mL solution of C16-DABCO dendrimer (33.3 μM), and inoculated with *S. aureus*, no visible biofilms were present after three days of incubation [43].

With the aim of reducing the incidence of implant-associated infections, severe post-surgery complications leading to patient disability and morbidity, and even death, Wang et al., (2011), for the first time, successfully coated three types of titanium-based substrates with PAMAM Ds modified with various percentages of poly(ethylene glycol) (PEG) to limit cytotoxicity [44]. Based on their inner and peripheral high content of amine groups, electrostatic interactions were more favored with bacterial membranes characterized by higher negative charge, causing their destruction. Consequently, the PAMAM coatings displayed greater bactericidal activity against Gram-negative *P. aeruginosa*, than against Gram-positive *S. aureus*, due to the higher negativity of their OM. The dendrimer adsorbed on titanium surfaces, proved durable bacteriophobic abilities, inhibited bacterial colonization even after 30 days and showed good stability and biocompatibility with human bone mesenchymal stem cells (hMSCs) [44].

In the same year, Navath and colleagues developed in situ forming PAMAM Ds-PEG biodegradable hydrogels, loaded with amoxicillin, and proposed these materials as novel agents for the delivery and sustained release of that specific antibiotic in the cervicovaginal region, in order to treat ascending genital infections [17]. Briefly, the PAMAMs-based amoxicillin formulations were prepared by the cross-linking of [(NH_2_)49-G4-(NH-PDP)15] dendrimer and 8-arm-PEG via formation of disulfide bridges and were tested both in vitro and in vivo. Amoxicillin release from these hydrogels (3%, 6% and 10% *w*/*v*) was sustained for more than 240 h, while in vivo assessment of the hydrogels, carried out using a pregnant guinea pig model, showed excellent tolerability, lack of alterations in vaginal pH and/or erythema up to 72 h and slow degradation. The gels were retained in the maternal tissues without being transferred across the fetal membranes [17].

Recently, another approach has been reported to obtain a synergistic action between cationic Ds and antibiotics [16]. Instead of binding the antibiotic to the dendrimer backbone, thus making a device that possesses both the intrinsic antimicrobial activity of the carrier and that of the antibiotic, synthetic polycationic and polyanionic Ds were administered in combination with levofloxacin (LVFX), in order to reduce the development of resistance to the drug, decrease its dosage and toxicity, as well as the environmental pollution resulting from the wide spread use of fluoroquinolones [16]. The germicidal activity of the combination made was investigated by treating *E. coli* ATCC 25922, *Proteus hauseri* ATCC 15442, and *S. aureus* ATCC 6538 strains for 24 h with fixed concentrations of LVFX and Ds. Satisfactory results, in terms of antimicrobial activity, were obtained only against the Gram-negative *E. coli* by the third-generation PPI-based cationic dendrimer, the only one of interest for this review, characterized by a dense maltose shell (PPI-G3-DS-Ma) and known to be endowed with a dose-dependent cytotoxicity, [16]. In fact, a combination of the dendrimer PPI-G3-DS-Ma (10 μM) and LVFX (0.01 μg/mL) reduced the growth of *E. coli* by 80%, while dendrimer or antibiotic alone reduced the same growth by only 6% or 4%, respectively [16]. This strategy, even if limited to *E. coli*, has allowed to significantly reduce the LVFX dosage, using the cationic dendrimer at a dose which, for a similar dendrimer of higher generation previously reported [63], showed very limited cytotoxicity on several human cell lines (cell viability around 80–100%) [63].

In this last study, in fact, the preparation and evaluation of the germicidal activity of an unmodified fourth-generation PPI D (PPI-G4) and of PPI Ds modified with 25% and 100% maltose attached to their surface (PPI-25%mG4 and PPI-100%mG4, respectively), was reported. The target bacteria were both Gram-positive (*S. aureus* ATCC 6538 and *S. epidermidis* ATCC 12228) and Gram-negative (*E. coli* ATCC 25922 and *P. aeruginosa* ATCC 15442) strains [63]. In parallel, the cytotoxic effects of all tested Ds were checked on a Chinese hamster fibroblast cell line (B14), a human liver hepatocellular carcinoma cell line (HepG2), a mouse neuroblastoma cell line (N2a) and a rat liver cell line (BRL-3A). The obtained results established that Gram-negative bacteria were refractory to the Ds, also at very high concentrations (100 μM). The unmodified PPI-G4 was the most active dendrimer, killing 90% of *S. epidermidis* isolates at 30 μM and 60% of *S. aureus* strains at 30 μM [63]. Unfortunately, due to its IC_50_ values in the range of 3.18–6.91 μM against the eukaryotic cells analyzed, its use for clinical applications must be excluded. PPI-100%mG4 was practically inactive against all the tested pathogens, while PPI-25%mG4 proved good activity against *S. aureus*, at concentrations that allowed high viability maintenance of all eukaryotic cell lines essayed [63].

Ds, due to a high density of exterior functional groups available for nitric oxide (NO) loading, were particularly attractive for preparing (NO)-releasing Ds characterized by large NO payloads and antibacterial activities against a wide range of pathogenic bacteria, including *P. aeruginosa* and *S. aureus.* The antibacterial activity of a library of (NO)-releasing PPI Ds was evaluated against Gram-positive and Gram-negative isolates, including methicillin-resistant *S. aureus* (MRSA) by Sun et al. (2012) [64] and it was compared with that of analogous PPI Ds, non-NO-releasing, taken as control. From the results, the NO-releasing Ds displayed both enhanced biocidal action directly proportional to Ds size (in terms of MW) and reduced toxicity against mammalian fibroblast cells. In addition, minimal toxicity against fibroblasts and the strongest biocidal activity (≥99.999% killing) was exerted by NO-releasing PPI Ds modified with styrene oxide (SO). The MBC values for each bacterial strain were determined in a non-conventional way, i.e., as the concentration of Ds that caused either a 3 or 5 log reduction in cell viability (compared to untreated cells for a particular bacterial strain) after 2 h. The best results obtained have been reported in Table 5 [64]. Concerning the cytotoxic effects against L929 fibroblasts cell lines, a very low cell viability (<20%) was observed when cells were exposed to concentrations of all G5-PPI Ds necessary to kill bacteria. On the other hand, both G2-PPI-PO and G2-PPI-NH_2_ exhibited minimal toxicity to fibroblast cells when tested at bactericidal concentrations. When administered at concentrations 2× MBCs, G2-PEG- and SO-modified Ds (namely 3 and 5) still inhibited fibroblast proliferation by approximately 76% and 64% respectively, therefore showing considerable cytotoxicity. Significantly, the toxicity of NO-releasing Ds against L929 fibroblasts was minimal (≥60% viability) when the devices were administered at both the minimum and twice the minimum concentration required to elicit 3-log killing against all tested bacterial strains. In particular, both G2 and G5 NO-releasing PPI-SO Ds, and their precursors, proved to be non-toxic to the fibroblast cells (>80% viability) even at concentrations necessary to induce 5-log killing against all species studied (10 and 1 μM) [64].

A library of amphiphilic NO-releasing PAMAM Ds were synthesized by Lu et al. (2013), through a ring-opening reaction between terminal primary amines present on the dendrimer and propylene oxide (PO), 1,2-epoxy-9-decene (ED) or different ratios of the two, followed by reaction with NO, in order to achieve the NO-releasing derivatives [65]. The hydrophobicity of non-NO-releasing and of NO-releasing Ds was tuned by varying the ratio of the external functionalization PO/ED. The bactericidal efficacy of the NO-releasing vehicles obtained was evaluated against planktonic Gram-negative *P. aeruginosa* strains and established *P. aeruginosa* biofilm. The results were correlated with the Ds exterior hydrophobicity (i.e., ratio of PO/ED), their size (i.e., generation), NO release and was compared to that one of non-NO-releasing compounds with analogous composition, taken as controls. It was observed that both the size and the exterior functionalization of Ds were pivotal factors influencing several parameters affecting the antibacterial activity such as dendrimer-bacteria electrostatic interactions and the capability to disrupt bacterial membranes, the efficacy to deliver NO, their migration within the biofilm, and their toxicity against mammalian cells [65]. All the NO-releasing Ds analyzed displayed higher antimicrobial activity than the non-NO-releasing Ds (Table 5) [65]. In more details, the first-generation (G1) PO-Ds were ineffective both against planktonic cells and biofilm cells of *P. aeruginosa*, while the G1 ED-Ds proved to have a potent antimicrobial activity against the same planktonic cells (MBC = 5 μg/mL), associated with low toxicity against L929 mouse fibroblast cells, and good activity against the biofilm cells (MBC = 15 μg/mL), unfortunately associated with considerable toxicity to L929 mouse fibroblast cells (cell viability < 40%). The optimal PO to ED ratios for biofilm eradication with minimal toxicity against L929 mouse fibroblast cells were 5:5 (MBC = 20 μg/mL, cell viability in the range 90–100%). The study by Lu et al. [65] demonstrated the importance of both the D size and its external functionalization (PO/ED) in determining the efficacy of the devices against established biofilms without compromising their biocompatibility with mammalian cells.

In the field of NO-releasing Ds, Worley et al. (2014) [66] described the synthesis of NO-releasing quaternary ammonium (QA)-functionalized generation 1 (G1) and generation 4 (G4) PAMAM Ds, where QA groups were supported by different alkyl chain lengths (i.e., methyl, butyl, octyl, dodecyl) via a ring-opening reaction. The secondary amines resulting from these reactions were further modified with *N*-diazeniumdiolate NO donors to achieve the NO-releasing QA-modified PAMAM Ds, capable of spontaneous NO release. Differently from non-NO-releasing QA-modified PAMAM Ds, the NO-releasing ones can exert an antibacterial dual action. In detail, the highly cationic structure of the macromolecules can inactivate the pathogens by disrupting their membranes, furthermore the release of NO can disarm bacteria, by causing oxidative and nitrosative stresses, by promoting the production of reactive NO byproducts [(N_2_O_3_), peroxynitrite (ONOO−)], by causing membrane destruction via peroxynitrite-induced lipid peroxidation and by triggering protein *S*-nitrosation and DNA deamination. In this regard, the antibacterial activity of the NO-releasing QA-modified PAMAM Ds was evaluated against the Gram-positive *S. aureus* and the Gram-negative *P. aeruginosa* species after four hours of exposure. Overall, the bactericidal activity of the devices was found to be influenced by dendrimer generation, QA alkyl chain length, and bacterial Gram class. The presence of shorter alkyl chains conferred an increased bactericidal activity, particularly against *P. aeruginosa*, for both generations, with NO-releasing Ds resulting markedly more potent in killing bacteria (Table 5). The toxicity of non-NO-releasing Ds and of NO-releasing Ds on L929 fibroblasts was minimal (about 60–100% and 80–110% viability respectively) when administered at the MBC concentrations, but a lower toxicity was observed for NO-releasing Ds, if compared to the non-NO-releasing ones [66].

NO-releasing scaffolds, including Ds, were developed by Backlund et al. (2014) and were evaluated as alternatives to current treatments for periodontitis (e.g., scaling/root planning and chlorhexidine), which are affected by limited efficacy, since they fail to suppress microbial biofilms satisfactorily over time, and since the use of adjunctive antimicrobials, in those conditions, can promote the emergence of antibiotic-resistant organisms [67]. The engineered NO-releasing G1-PAMAM-PO-Ds demonstrated a 3-log reduction in the growth of periodontal pathogenic bacteria such as *Aggregatibacter actinomycetemcomitans* and *Porphyromonas gingivalis* (MBC = 2000 and 1000 μg/mL). *Streptococcus mutans* and *Streptococcus sanguinis*, typical caries-associated organisms, were, on the contrary, substantially less sensitive to NO treatment (MBC = 48,000 μg/mL). NO-releasing Ds showed less toxicity against human gingival fibroblasts at concentrations needed to eradicate periodontal pathogens than clinical chlorhexidine concentrations [67]. Although the authors considered these results promising and suggested NO-release dendrimer scaffolds as novel platforms for the development of periodontal disease therapeutics, in our opinion, the MBC values observed are too high to make these structures usable in clinical practice.

Later, Backlund and colleagues prepared generation 1 (G1) propyl-, butyl-, hexyl-, octyl-, and dodecyl-functionalized PAMAM-D *core* scaffolds, subsequently converted to N-diazeniumdiolate NO donors. The killing effect of hydrophobic G1 NO-releasing Ds on *S. mutans* and their capability to disperse its biofilm were examined at pH 7.4 and 6.4, the latter being a value promoting the dental caries process [68]. The bactericidal action of the NO-releasing Ds against both planktonic and biofilm-producing strains of *S. mutans* was shown to be greater by increasing the alkyl chain length and lowering the pH (pH = 6.4). The authors hypothesized that the improvement in bactericidal efficacy at pH 6.4 was attributed to both increased surface positive charge of the scaffold, responsible for a better dendrimer-bacteria interaction, resulting in membrane damage, and to a faster NO-release kinetics from proton-labile *N*-diazeniumdiolate NO donors. Specifically, only octyl- and dodecyl-modified PAMAM Ds were actually effective in eradicating planktonic *S. mutans* cells (MBC values in the range of 12–25 μg/mL), but were ineffective in eradicating *S. mutans* biofilm cells (MBC values in the range of 1000–2000 μg/mL). In addition, even if NO-release Ds showed mitigated cytotoxicity on HGF-1 human gingival fibroblasts, compared with non-NO-releasing Ds, the viability of the cells at the concentrations required for biofilm eradication was in the range of 15–20%. These findings established that octyl- and dodecyl-modified PAMAM Ds can be taken into consideration for clinical application with the aim of killing planktonic forms of *S. mutans* strains, but too toxic to counteract its biofilm structure [68].

The following year, alkyl-modified NO-releasing PAMAM Ds, from first to fourth generation, comprising butyl and hexyl chains, and non-NO-releasing analogs, similar to the PAMAM Ds developed by Backlund et al. (2014) [68], were prepared by Worley et al. [69]. They were evaluated for their bactericidal activity within 24 h against biofilms produced by Gram-positive and Gram-negative bacteria, including antibiotic-resistant strains, assuming that the antibiofilm abilities of the alkyl chain modified NO-releasing and non-NO-releasing Ds could be improved by increasing the size of the Ds and the density of the functional groups [69]. The results showed that the antibiofilm action of the Ds was dependent on their generation, the bacterial species, and the length of the alkyl chain of the devices. In particular, the more effective eradication of the biofilm was associated with the greater infiltration capacity of the Ds into the biofilm biomass. In this regard, the introduction of the ability to release NO has significantly improved the antibiofilm activity of those Ds incapable of effective biofilm penetration. In particular, G1 butyl Ds were practically ineffective against planktonic cells of *P. aeruginosa*, *S. aureus* and against MRSA strains, while G1/G4-butyl/hexyl non-NO-releasing Ds and the analogous NO-releasing devices showed, against *P. aeruginosa*, MBC values in the range of 10–50 μg/mL and of 25–50 μg/mL respectively. The G3-hexyl non-NO-releasing D was recorded as the most active one (MBC value = 10 μg/mL). The same non-NO-releasing and NO-releasing Ds that showed activity against *P. aeruginosa* were practically ineffective against *S. aureus* and MRSA strains, with the most active one being the G3-hexyl non-NO-releasing D, which showed an MBC value of 50 μg/mL, against both *S. aureus* and MRSA strains [69]. Minimum biofilm eradication concentrations after 24 h of exposure (MBEC_24h_) against biofilm produced by *P. aeruginosa*, *S. aureus*, and MRSA isolates were very high for all the tested Ds and, although in general the NO-releasing Ds were better than the non-NO-releasing ones, the most active Ds were the G3 hexyl (MBEC_24h_ = 200, 100, 100 μg/mL against the three species respectively) and the G3 hexyl/NO ones (MBEC_24h_ = 100, 100, 100 μg/mL against the three species respectively) [69]. Interestingly, the best performant G3 hexyl Ds, against both planktonic cells and biofilm, showed in in vitro cytotoxicity essay on L929 mouse fibroblasts, an IC_50_ of 450 μg/mL, either for non-NO-releasing and for NO-releasing compounds, thus establishing a low level of cytotoxicity at both MBC and MBEC_24h_ concentrations.

In the search for new alternatives to overcome bacterial drug resistance, an appealing option could be the application of bacteriophage enzymes, such as endolysins, which are able to degrade bacterial peptidoglycan (PG), and to lead to bacterial cell lysis. Starting from this approach, and in order to help endolysins to gain access to PG, Ciepluch et al. (2019) recently combined the endolysins produced by phage KP27 with unmodified and peripheral modified 20% maltose containing PPI Ds, known for their capacity to destabilize the bacterial OM [70]. The antibacterial activities of mixtures containing the modified PPI Ds and the endolysins (12 μM) in a molar ratio of 1/1 and ¼, were essayed against *P. aeruginosa* PAO1 (ATCC 15692) wild-type and mutant strains with reduced antigen O in LPS. The results were compared to those obtained by the administration of endolysins 12 μM or PPI Ds (12 μM and 50 μM), administered separately. The findings showed that the co-administration of endolysins and maltose-modified PPI D in ratio of 1/1 (the combination used contained a concentration of the dendrimer not toxic to eukaryotic cells), significantly enhanced the antibacterial activity of the enzyme (40–50%) against *P. aeruginosa* PAO1 and WAAL strains [70].

By divergent growth method, Gholami et al. (2017) synthesized a high MW (116,493 g/mol) G7 PAMAM-D owing 512 positively charged amine groups, thus making the most potent broad-spectrum CAD synthesized to date, endowed with low levels of cytotoxicity on two types of eukaryotic cell lines [50]. Its antibacterial behavior was evaluated on several species such as *P. aeruginosa* (n = 15), *E. coli* (n = 15), *A. baumannii* (n = 15), *S. dysenteriae* (n = 15), *K. pneumoniae* (n = 10), *P. mirabilis* (n = 15), *S. aureus* (n = 15) and *B. subtilis* (n = 10). Additionally, representative standard strains for each species were included. As reported in Table 5, G7-PAMAM-D inhibited the growth of both Gram-positive and Gram-negative pathogens, displaying MIC_50_ and MIC_90_ values in the range of 2–4 μg/mL and 4–8 μg/mL, respectively, corresponding to very low micromolar concentrations (0.017–0.034 and 0.034–0.068 μM, respectively) [50]. MBC_50_ and MBC_90_ values were found to be 64–256 μg/mL and 128–256 μg/mL respectively, corresponding to micromolar concentrations of 0.55–2.2 and 1.1–2.2 μM, respectively (Table 5) [50]. The cytotoxicity of G7-PAMAM was evaluated by MTT assay on human intestinal cancer cell lines HCT116 and NIH 3 T3, observing a reduction in viability of 44.6% and 43% respectively at the highest concentration tested (0.85 μM), after 72 h of exposition (Table 5) [50]. It should be kept in mind that cytotoxicity data do not provide information concerning hemolytic toxicity, deriving from a non-specific disruptive action on RBC membranes, which is one of the biggest concerns associated with the possible clinical use of highly cationic and typically membrane-active antimicrobial agents. Based on such considerations, further investigations on the hemolytic toxicity of G7-PAMAM are mandatory to consider it as a promising antimicrobial alternative to conventional antibiotics.

To achieve antimicrobial coatings able to prevent bacterial adhesion, the Zhan’s group (2015) successfully prepared hyaluronic acid/PAMAM dendrimer (HA/PAMAM-D) multilayers on a poly(3-hydroxybutyrate-co-4-hydroxybutyrate) [P(3HB-4HB)] substrate by a layer-by-layer self-assembly method [71]. By using QCM-D, the authors showed that both the HA outer layer and the PAMAM-D outer layer revealed anti-adhesion activity against *E. coli*. Otherwise, by using a live/dead assay, it was observed that while the PAMAM-D outer layer could also exhibit bactericidal activity against *E. coli*, the outer layer of HA had no such activity. HA/PAMAM-D coatings were able to maintain the antibacterial and anti-adhesion activity after storage in phosphate-buffered saline for up to 14 days. Moreover, the in vitro MTT assay showed that the multilayers were not cytotoxic against L929 cells, and that HA molecules in the multilayers could also improve the biocompatibility of the film [71].

In previous studies, with the aim of obtaining a bacteriophobic coating to be applied on cotton fabric, cationic PAMAM-chitosan (CTS) dendrimers (PAMAM-CTS-Ds) were prepared by Klaykruayat et al. (2010) [72]. Briefly, a quaternary ammonium hyperbranched dendritic PAMAM of generation 2.5, customized by post-synthesis methylation of a methyl ester terminated PAMAM-D, was employed to modify flake CTS, and the cationic PAMAM-CTS-D obtained was applied to cotton fabric at 1% *w*/*w*, using a padding method. The antimicrobial performance of the newly modified fabric was assessed against *S. aureus* and compared to a similar fabric obtained by applying unmodified CTS. Furthermore, the antimicrobial activity of not neutralized CTS films and of cationic PAMAM-CTS-D films was also evaluated [72]. The antimicrobial evaluations were preliminarily carried out in terms of visual detection and reduction of the percentage of microbial growth, using a single application dose. The 1% *w*/*w* CTS-impregnated fabric, treated with NaNO_2_, according to a necessary procedure, showed no antimicrobial activity, thus indicating that CTS itself was not sufficient to inhibit the growth of *S. aureus*, when applied on fabric [72]. Nonetheless, native CTS film (0.33 g) showed an excellent antimicrobial activity (100% reduction of the growth of *S. aureus*), due to the positively charged CTS amine groups. The cationic PAMAM-CTS-D film (0.25 g), also exhibited a strong antimicrobial activity against *S. aureus* (99.99% reduction), comparable to that of the native CTS film. Significantly, in contrast to cotton fabric coated with CTS alone, cotton fabric treated with cationic PAMAM-CTS-D (1% *w*/*w*) displayed an excellent antimicrobial activity (98.75% reduction of the growth of the pathogen) [72]. Although cationic PAMAM Ds could potentially enhance the performance of CTS as an antimicrobial coating for cotton fabric, probably due to their high cationic character, the preliminary findings highlighted by Klaykruayat et al. require further characterization and verification on other bacterial species. Similarly, more investigations are needed regarding the dose-dependence of the effect observed within the fabric, including its resistance to multiple wash cycles.

### 4.2. PEI-Based Dendrimers (PEI-Ds)

As far as our knowledge is concerned, the application of water-soluble PEI derivatives, containing quaternized ammonium salt groups with long alkyl or aromatic groups, such as antimicrobial polymers, and of water-insoluble hydrophobic modified PEIs, such as antimicrobial coatings have been widely documented [56]. On the contrary, the application of PEI-Ds as antimicrobial agents has been reported in a few studies, as observable in Figure 3. Table 6 reports the most representative of those documented.

Concerning their proposed mechanism of action, the protonated ammonium-terminated groups of the arms of PEI-Ds are the positive part serving for electrostatic interactions with bacteria membranes, while the non-protonated amine groups and ethylene backbone serve as hydrophobic groups, helpful for PEIs diffusion through bacterial membranes. Overall, the repeated cationic amphiphilic structures, along the dendrimer backbone of unmodified PEI-Ds, by providing the necessary cationic amphiphilic structures, mimic of CAMPs, are capable of inducing membrane permeabilization, disruption, and bacterial death. In this regard, a systematic investigation of the antimicrobial activity and toxicity of PEI-Ds which differ in generation number and molecular weight has been reported by Gibney and colleagues (2012) [56]. The authors focused on the structure-activity relationship responsible for antimicrobial activity against *E. coli* and *S. aureus* as well as on the toxicity against human red blood cells (hemolysis) and human epithelial carcinoma HEp-2 cells. The polymer-induced permeabilization of bacterial cell membranes of *E. coli* and *S. aureus* was also evaluated. The PEI-Ds under study exhibited considerable antimicrobial activity (MIC values = 16–32 μg/mL) and selectivity against *S. aureus*, whereas they demonstrated poor activity against *E. coli*, with MIC values up to >1000 μg/mL). All compounds proved low hemolytic toxicity, while considerable cytotoxicity was observed mainly for PEI-Ds with high MW and after 24 h of exposure.

Lately, considering the wide range of applications of oxadiazole compounds in biomedicine and the multivalent PEI-Ds, which provide many branches such as –NH_2_ functional groups exploitable for modifications with several bioactive heterocyclic derivatives, the synthesis of seven PEI-based oxidiazole-modified Ds (PEI-dend-4[N[(Ts)(2-(methyl)-5-aryl-1,3,4 oxadiazole)]] was reported [55]. Prepared from PEIs and 2-aryl-1,3,4 oxadiazole derivatives differently substituted on the phenyl group, the achieved Ds were investigated for their in vitro antimicrobial activities by MICs (μg/mL) determination. Results indicated that although four compounds manifested from moderate to very poor antimicrobial activity, two of them exhibited broad-spectrum antimicrobial activity against bacteria and fungi, highlighting that the presence of more electron donating groups at *para* position in phenyl ring bearing oxadiazole influences antimicrobial activity positively [55].

### 4.3. Cationic Peptides Dendrimers

Natural CAMPs such as polymyxins or gramicidins, known to display high antimicrobial potency unfortunately associate with remarkable cytotoxicity to the host cells, have inspired the synthesis of cationic antimicrobial peptides. In this regard, cationic peptides with repeating sequences of arginine and tryptophan, (RW)n, have highlighted that short chains of R and W comprise a pharmacophore for mimicking the antimicrobial activity of the natural CAMPs [58]. Studies have demonstrated that the presence of multivalent dipeptides or tetrapeptides on different scaffolds, further enhances the antibacterial effects. In this context, since multivalence is one of the nonpareil properties of Ds, six new Ds, on which different dipeptide combinations of cationic and hydrophobic amino acids are linked to a four-branched lysine dendritic *core* were constructed by Young et al. (2011) [58]. In addition to RW, these dipeptides include RF, RY, KW, KF, KY, and HW sequences. These materials were compared with the correspondent linear and polymeric materials for their antimicrobial activity against both Gram-negative *E. coli* and *A. baumannii* species, as well as towards the Gram-positive *S. aureus* and for their hemolytic toxicity. Interestingly, more than one dendrimer peptide proved to possess antimicrobial activity higher than that of both linear and polymeric correspondent peptides and lower hemolytic toxicity. In particular, the better performant dendrimer peptide, named (RW)4D, provided IC_50_ values (concentration of the agent that inhibit 50% of bacterial growth) of 3.9, 15 and 42 μg/mL against *E. coli*, *A. Baumannii* and *S. aureus* respectively, associated with a very low hemolytic toxicity, since the hemolytic cytotoxicity (HC_50_) value (concentration of agent that lyses 50% of RBC) was 1962 μg/mL, as reported in Table 7. In addition, the results of the study established that even extended exposure to sub-lethal doses of (RW)4D elicited much lower levels of resistance than traditional antibiotics or antimicrobials such as ciprofloxacin, vancomycin, chlorhexidine and gentamicin in multidrug-resistant strains [58].

To enhance the antibacterial activity and the half-life of melectin (MEP, GFLSILKKVLPKVMAHMK-NH_2_), which exhibited high antimicrobial activity against Gram-positive and Gram-negative bacteria at MIC values from 4 to 120-times lower than the HC_50_ value, Niederhafner et al. (2010) reported its dendrimerization [73]. In the study, 23 dendrimer derivatives of melectin have been synthetized and evaluated for their antimicrobial activity against *B. subtilis*, *S. aureus*, *E. coli* and *P. aeruginosa* species as well as for their hemolytic toxicity. According to the results, in contrast with the findings reported by Young et al. [58], except for some cases, in which HC_50_ increased and therefore the hemolytic activity reduced, it was generally superior [73]. On the other hand, their antibacterial activity against *B. subtilis*, *E. coli* and *P. aeruginosa* was generally improved and anyway some very appealing materials proved very favorable HC_50_/MICs ratios [73]. Curiously, the antibacterial activity against *S. aureus*, except for three cases, decreased among the dendrimer derivatives. Collectively, as observable in Table 6, the ranges of both MICs and HC_50_ values are very wide, depending on the levels of multivalence and some modification in the peptide sequence made up by the authors, affecting the cationic character of the prepared peptide Ds.

Later, the in vitro activity of the third-generation antimicrobial peptide dendrimer containing the dipeptide sequences KL, named G3KL, was evaluated against 32 *A. baumannii* strains, including 10 OXA-23, 7 OXA-24, and 11 OXA-58 carbapenemase producers isolates and against 35 *P. aeruginosa* strains, including 18 VIM and 3 IMP carbapenemase producers isolates and the results were compared to the activities of standard antibiotics [74]. The peptide dendrimer showed MICs_50/90_ values of 8/8 μg/mL and MBCs_50/90_ values of 8/8 μg/mL against both species collections, minimal hemolytic concentration of 840 μg/mL vs. 2000 μg/mL for polymyxin B, and stability in human serum, being its half-life [t_1/2_] of 18 h [74].

A series of eight amphiphilic peptide Ds, built up around a dendronized ornithine (Orn) *core*, were synthesized by Polcyn et al. (2013) and evaluated for their antimicrobial properties against *S. aureus* ATCC 25923, *S. aureus* ATCC 43300, *E. coli* ATCC 25922, and *P. aeruginosa* ATCC 27853 strains [75]. As can be observed in Table 6, the achieved peptide Ds showed very different MIC values: two Ds proved low activity against all the bacteria, four Ds proved appreciable activity only against *S. aureus* ATCC 25923 strain, and the residual two manifested good activity against all the bacteria, with mild effects against *P. aeruginosa*. A higher antimicrobial potency was correlated with a higher charge density and branching and to a higher lipophilicity of the residues located at the C-terminus. In particular, the most efficient peptide Ds were the isomeric hexachlorides, namely 3d and 3h, whose structure presented a C12 lipophilic chain, due to the dodecylamine residue. Of the two structures, dendrimer 3d, which possesses cationic amine groups less cluttered and free to interact electrostatically with the bacterial membrane, has shown MIC values of 4, 0.99, 4, and 16.9 μg/mL respectively, against *S. aureus* ATCC 25923, *S. aureus* ATCC 43300, *E. coli* ATCC 25922 AND *P. aeruginosa* ATCC 27853 strains. Experiments on hemolytic toxicity on RBCs showed a strict correlation with the antimicrobial activity of the Ds and their structural properties. In fact, the compounds that were practically inactive against bacteria, were also the less toxic. Unexpectedly, compound 3d, presenting the best performance in terms of antibacterial activity, caused only about 35% of hemolysis when used at 100 μM concentration [75].

A series of peptide Ds were synthetized by Stach et al. (2012) and investigated for their antimicrobial activity against *B. subtilis*, *E. coli* and *P. aeruginosa* species and for their hemolytic toxicity [76]. Although some compounds were totally inactive and a limited activity was generically observed against *P. aeruginosa*, the compound (H1), with sequence Leu8(Lys-Leu)4(Lys-Phe)2Lys-LysNH_2_ (Lys = branching lysine) and the analog compound (bH1), with sequence Leu8(Dap-Leu)4(Dap-Phe)2Dap-LysNH2 (Dap = branching 2,3-diaminopropanoic acid) showed good antimicrobial activities, mediated by membrane disruption. In particular, bH1 was the most potent as an antimicrobial, and more active than Indolicidin (ILPWKWPWWPWRRNH_2_) against all the bacteria tested. Moreover, it showed a good antibacterial activity also against *P. aeruginosa* (MIC values = 20 μg/mL). In addition, bH1 displayed very low hemolytic activity, with a minimal hemolytic cytotoxicity (MHC) ≥ 2000 μg/mL [76]. H1 and bH1 were also tested against strains of *S. aureus*, *S. epidermidis* ATCC 14990, *En. faecium* and *E. coli* species. The results showed MIC values in the range 8→32 μg/mL and in some cases their activity was higher than that of Polymixin B and/or Ampicillin [76].

Scorciapino et al. (2012) prepared a semisynthetic dendrimer (dimeric) peptide (SB056), by alternating hydrophilic and hydrophobic amino acids, achieving a membrane-active peptide able to form amphiphilic β-strands in a lipid environment [77]. Lipid monolayer surface pressure experiments revealed that SB056 exerted its membranolytic activity by a sort of lipid-induced aggregation mechanism. SB056 showed high activity against multidrug-resistant Gram-negative bacteria comparable to that of colistin and polymyxin B, with an even broader spectrum of activity than numerous other reference compounds. On the contrary, its activity on Gram-positive bacteria was more limited [77] (Table 5).

In a subsequent study, investigations into the antibiofilm activity and hemolytic toxicity of SB056 were carried out [77]. According to the results, the peptide dendrimer provided a HC_50_ value of 159 μg/mL, thus asserting a low level of hemolytic toxicity and its potentials as a therapeutic alternative to conventional antibiotics to treat infections by multidrug-resistant bacteria. With regard to its antibiofilm activity, the antimicrobial activity of SB056 was first investigated against planktonic forms of reference strains of *S. epidermidis* and *P. aeruginosa* under biofilm-like conditions, unfortunately obtaining very high MIC values (≥102.4 μg/mL). Then, the ability of the peptide to inhibit biofilm formation was evaluated against the same strains. When tested at 25.6 μg/mL on *S. epidermidis*, SB056 exhibited strong antibiofilm activity, causing a reduction of approximately 98% of the biofilm biomass. When it was tested against *P. aeruginosa*, SB056 caused a 90% reduction of biofilm biomass and only at the concentration of 51.2 μg/mL.

Later, the same group amplified their studies concerning the SB056 dendrimer peptide (in this new study named den-SB056), enhancing the amphiphilic profile of the original sequence [WKKIRVRLSA-NH_2_] by interchanging the first two residues [KWKIRVRLSA-NH_2_] and obtaining den-SB056-1 [78]. Results obtained against *E. coli* and *S. aureus* planktonic strains confirmed a reduced activity on *E. coli* (MICs = 16 μg/mL vs. 8 μg/mL of the native dendrimer) and an unchanged activity on *S. aureus* (MICs = 32 μg/mL) associated with increased hemolytic toxicity (HC_50_ = 87 μg/mL). Concerning bactericidal activity against the same strains, the modified dendrimer peptide was less active on *E. coli* (MBC = 16 vs. 8 μg/mL), whereas it was more active on *S. aureus* (MBC = 32 vs. 64 μg/mL) [78]. Nonetheless, the efficacy of den-SB056-1 was higher than that of den-SB056 both against Gram-positive and Gram-negative bacteria, specifically when tested in broths supplemented with physiological concentration of electrolytes [78]. Results obtained from the evaluation of den-SB056-1 activity on sessile forms of *S. epidermidis* and *P. aeruginosa* bacteria established a remarkable reduced activity in decreasing biofilm biomass, especially if compared to the same activity possessed by the native den-SB056 [77].

Recently, the group of Scorciapino, reconsidered the small peptide dendrimer (den-SB056) and its more hydrophobic modified derivative (den-SB056-1). Aiming to increase their multivalence, two copies of den-SB056 or of den-SB056-1 were bound to the α- and ε-nitrogen atoms of one lysine *core*. Finally, an 8-aminooctanamide residue was added at the C-terminus of the lysine to improve its cationic character and its membrane affinity [79].

A second-generation peptide dendrimer (2D-24), containing residues of lysine and tryptophan, was synthetized by Bahar et al. (2015) by solid phase method and its bioactivity was investigated against planktonic, biofilm, and persister cells of the wild-type *P. aeruginosa* (PAO1) and its mucoid mutant strain (PDO300) [14]. 2D-24 was found to definitely kill planktonic cells of both strain types at concentrations of 77.5 μg/mL and to kill 94% of their biofilm cells at concentrations of 46.5 μg/mL, suggesting that 2D-24 is able to penetrate the biofilm matrix and the alginate layer of the mucoid strain. Concentrations up to 310 μg/mL of 2D-24 were necessary to kill 69 and 89% of multidrug tolerant persister cells of PAO1 and PDO300, respectively. Although such a high concentration establishes a total inactivity of the dendrimer against persister cells, the synthetic peptide proved a promising synergistic effect when administered in combinations with ciprofloxacin, tobramycin, or carbenicillin. Based on hemolysis assays, using sheep erythrocytes and on cytotoxicity tests, using a coculture model of PAO1 and IB3-1 human epithelial cells, 2D-24 exhibited very low levels of hemolytic toxicity (HC_50_ > 1000 μg/mL) and was found to kill *P. aeruginosa* cells at concentrations that are not toxic to mammalian cells (25 μg/mL) [14].

Stach and colleagues (2014), inspired by the prevalence of leucine and lysine residues in several natural CAMPs, prepared six third-generation *L*-peptide Ds, namely G3KL, G3LK, G3KK1, G3KK2, G3LL1 and G3LL2 by solid phase peptide synthesis (SPPS) [80]. The investigation results on their potency against *P. aeruginosa*, *E. coli*, and *B. subtilis* established that G3KL, with the sequence (KL)8(KKL)4(KKL)2KKL, showed a good broad spectrum of activity, with high selectivity for bacteria cells and low hemolytic properties. Further investigations on dendrimer G3KL and on its enantiomer (DG3KL) were performed in the presence of human serum, which represents a degradation factor for natural CAMPs, thus limiting their activity in vivo. Both Ds proved high antimicrobial activity against *P. aeruginosa* PAO1, reporting MIC values of 2 μg/mL, with partial degradation (60% after 24 h) for DG3KL, and of 0.5 μg/mL, with no degradation after 24 h for DG3KL [78]. G3KL and DG3KL were also tested against four clinical isolates of *P. aeruginosa*, resistant to at least two different classes of antibiotics, including β-lactams, aminoglycosides or quinolones, and also against several reference strains, such as *A. baumannii* ATCC19606, *E. coli* W3110, and *E. aerogenes* 13048 and against lipopolysaccharide (LPS) mutant strains of *P. aeruginosa*. According to the results, the peptide Ds showed MIC values in the range 2–32 μg/mL on all resistant strains of *P. aeruginosa* and reference strains of *A. baumannii* and *E. coli* examined [80]. They showed lower activity on *E. aerogenes* (MIC values = 64 and 32 μg/mL, for the two Ds respectively), but comparable activity against LPS mutant strains of *P. aeruginosa* (Table 6) [80].

Recently, the *L*-peptide dendrimer G3KL was additionally taken into consideration by the group of Siriwardena [81]. By using virtual screening techniques in the field of Ds and a chemical-space-guided approach for the first time to search for new and improved analogs of G3KL, from a very large virtual library of compounds, Siriwardena et al. (2018) identified the dendrimer peptide T7 as the most improved one. T7 showed significant action across all the strains tested, through a membrane disruption mechanism, combined with excellent stability in the presence of serum, and negligible hemolytic activity (Table 6). T7 proved expanded activity range against Gram-negative pathogenic bacteria including *K. pneumoniae* and promising activity in an in vivo infection model by a multidrug-resistant strain of *A. baumannii* [81]. Due to their studies, the authors established that dendrimer size does not limit the activity of dendrimer peptides.

The following year, the Siriwardena group [82] obtained a new peptide dendrimer (DC5) including in its structure both the outer branches of a peptide dendrimer active against *P. aeruginosa*, *A. baumannii* and methicillin-resistant *S. aureus* (TNS18), and the *core* of T7, previously reported, and active against *P. aeruginosa*, *A. baumannii* and *K. pneumoniae*. DC5 was able to display the activity features of both parent compounds and retained a similar mechanism of action [82]. In particular, DC5 exerted antimicrobial activity against PAO1 at very low MIC values, both in the presence or in the absence of human serum (MIC values = 2–4 μg/mL) and presented a good activity against multi drugs resistant (MDR) strains of Gram-negative and Gram-positive bacteria (MIC values in the range of 4–32 μg/mL) and low levels of hemolytic toxicity (MHC = 500 μg/mL) (Table 6) [82].

Together with three linear peptides [(RW)n-NH_2_, where n is 2, 3, or 4], a dendrimer peptide, (RW)_4D_ was synthetized by Chen et al. (2011) [83]. The bactericidal activity of (RW)_4D_ was tested against planktonic persister cells, planktonic regular cells, and persister cells in preformed biofilms of the *E. coli* HM22 strain. Against planktonic persister cells, a dose-dependent killing activity was observed and significant killing was detected already at a concentration of 20 and 40 μM of the dendrimer peptide, which showed a reduction in the number of viable cells by a half log and one log, respectively [83]. Unexpectedly, the exposure of *E. coli* cells to a concentration of 80 μM of (RW)_4D_ only led to a decrease of 2-log in the killing activity. When normal *E. coli* cells were exposed to (RW)_4D_, a similar scenario was observed. Interestingly, when the bactericidal activity of (RW)_4D_ was evaluated on *E. coli* HM22 persister cells residing in 24 h sessile biofilms preformed on 316L stainless steel coupons, a potent dose-dependent dispersion of biofilm cells was detected [83]. In particular, after treatment for 1 h with concentrations of (RW)_4D_ equal to 20, 40 and 80 μM, the total number of viable biofilm cells were reduced by 77.1%, 98.8% and 99.3% respectively, compared to the untreated control. In addition, the same treatments reduced, the tolerance of the biofilm cells to ampicillin by 57.2%, 96.9% and 99.1% respectively. Furthermore, no viable cells were found after the biofilms were treated with concentrations of (RW)_4D_ of 40 or 80 μM, followed by 5 g/mL ofloxacin, suggesting that all persister cells were eliminated [83].

Interestingly, tissue attachment and biofilm formation due to *P. aeruginosa*, is mediated, among other factors, by two lectins, LecA and LecB, that bind specifically to galactosides and fucosides. Therefore, a successful approach to treat infections caused by *P. aeruginosa* and to inhibit biofilm formation and/or to disperse established biofilm, may include the treatments with lectin-binding saccharide solutions [84]. In this regard, the Kadam group reported the synthesis and the evaluation of the effectiveness of lectin inhibitors in the form of glycopeptide Ds, with four fucosides (FD2) [85] or four aromatic galactosides and aliphatic thiogalactosides (GalAG2 and GalBG2) [86] at the end of a common peptide dendrimer scaffold. Except for GalBG2, they were all able to bind tightly to their respective lectins and, consequently, to block formation and to induce dispersion of *P. aeruginosa* biofilms. Subsequently, Kadam and colleagues investigated several amino acid sequence variations for FD2 [87,88] and for GalAG2 [84], which were found to modulate the lectin-binding affinity, the solubility of the Ds, and the capability to inhibit biofilm formation.

More recently, Visini et al. (2015) reported structural studies concerning galactoside Ds of type GalA, capable of binding to *P. aeruginosa* LecA and showed that lectin aggregation is necessary for biofilm inhibition. At the same time, these studies have established the importance of the multivalence of glycopeptide Ds as a unique opportunity to control *P. aeruginosa* biofilms [89].

Based on the results obtained by Visini et al. in their studies [89], the year after, Bergman et al. (2016), prepared various G3 and G4 analogs of GalAG2 and GalBG2 using the multiple chloroacetyl cysteine (ClAc) thioether ligation as the key step. By using different approaches such as essays of inhibition of hemagglutination, isothermal titration calorimetry tests and biofilm inhibition evidence, the authors showed that G3 Ds bound LecA slightly better than their parent G2 Ds and induced complete inhibition and dispersal of biofilms of *P. aeruginosa*, while G4 Ds showed reduced binding and no biofilm inhibition capabilities [90].

That same year, Michaud, Visini, Bergaman et al., extended previous research on glycopeptide Ds, through further synthetic variations and activity combinations approaches, to obtain more potent biofilm inhibitors [91]. First, the multivalent chloroacetyl cysteine thioether ligation was used as an efficient method to build heteroglycoclusters including Het1G2-Het8G2 and (Het2G1-Cys)_2_ targeting both LecA and LecB. Subsequently, in a second approach, LecB targeting glycopeptide Ds, displaying analogs of the Lewis^a^ antigen (Le^a^CxG2) and β-fucoside derivative (FucC6G2), were investigated. In addition, to test the antimicrobial effects of the cationic peptide backbones contained in Het1G2-Het8G2 alone, non-glycosilated peptide Ds were prepared as controls. Heteroglycoclusters incorporating cationic residues displayed enhanced biofilm inhibition capability, associated with bactericidal behavior similar to that of membrane disrupting polycationic Ds. In particular, the best performing Het7G2 dendrimer showed an MBIC value of 13 μM, an MBC value of 13 μM and the ability to disperse 100% of the already established biofilms, at a concentration of 50 μM. Analogous Ds of the Lewis^a^ antigen, a natural LecB ligand, though stronger LecB ligands, displayed a decrease in biofilm inhibition, when compared with parent dendrimer FD2 (MBIC = 30 μM and approximately 88% dispersal at 50 μM). On the contrary, FuC6G2 showed a slightly better capacity to inhibit both biofilm formation (MBIC = 9 μM vs. 20 μM) and its dispersion (100% dispersal at 30 μM vs. 50 μM). However, from evaluations of non-glycosilated peptide Ds, it was observed that two Ds of this series (AcG2xK and NG2) were, one comparable with Het7G2 (AcG2xK) and the other the most active compound in terms of biofilm formation inhibition activity (MBIC = 13 and 2.5 μM) in killing bacteria under biofilm formation conditions (MBC = 13 and 2.5 μM) and in dispersing the established biofilm (100% dispersal at 50 μM for AcG2xK and at 22 μM). Finally, the authors proved that the synergistic application of the LecB specific non-bactericidal antibiofilm dendrimer FD2, and the antibiotic tobramycin (both compounds applied at sub-inhibitory concentrations), allowed effective biofilm inhibition and dispersal. In particular, strong synergistic effects were observed from the co-administration of FD2 with 0.1 μM Tobramycin, with new MBIC values of 5 μM, associated with a 100% dispersal of the established biofilm at a concentration of 10 μM [91].

### 4.4. Entirely Polyester-Based Uncharged Scaffolds Peripherally Modified with Cationic Amino Acids

Despite being considered very attractive for biomedical applications because of their lower density of cationic charge when compared to PAMAM Ds, PEI-Ds, and PPI Ds, generally associated with low cytotoxicity [92,93,94,95], only very recently uncharged dendrimer matrices, decorated with amino acid residues, were considered to be less toxic antimicrobial alternatives. Within this category, peripherally amino acid-modified, polyester-based dendrimer scaffolds, obtained starting from 2,2-*bis*(hydroxymethyl)propanoic acid (*bis*-HMPA), as AB2 monomer, have several features that make them particularly suitable as novel antimicrobial agents. These structures are characterized by good biodegradability, due to the easy physiological hydrolysis [92,93,94,95], a favorable hydrophilic lipophilic balance (HLB) due to the presence of an uncharged matrix and a cationic periphery, and can be esterified with a high number of amino acids, given the high multivalence of Ds. For these reasons, amino acid-modified polyester-based Ds could mimic synthetic dendrimer peptides which, as reported in the previous section, have provided remarkable results as antimicrobial agents, together with low levels of toxicity to eukaryotic cells. Research in this field is, however, still very limited and, as far as our knowledge is concerned, the literature presents only two studies. A further study not dealing with Ds but only dendrons, reporting the synthesis of low-generation polyester dendrons, based on *bis*-HMPA, peripherally cationic for the presence of β-alanine, has been recently published by Chen and colleagues (2020) [96] (Table 8).

In this regard, probably inspired by previous studies (2017) [92,93], hydroxyl functional Ds of one to five generations based on bis-HMPA and endowed with a tri-functional *core* corresponding to the International Union of Pure and Applied Chemistry (IUPAC) name of 2,2-bis-hydoxymethyl-1-butonol, and an analogous fourth-generation dendrimer, with a disulfide *core*, have been synthetized [18]. Both uncharged matrices were modified to bear from 6 to 96 peripheral amino groups, through esterification reactions with the non-natural amino acid β-alanine, achieving two types of cationic Ds which differ from one another in the inner *core* [18]. In particular, the achieved biodegradable Ds without a disulfide *core* (TMP-Gx-NH_3_^+^TFA) were tested for cytotoxicity against human skin cells (HDF), monocytes (RAW 264.7), and glioblastoma cells (U87) for 24 h with concentrations of 0.1–50 μM and with the extreme concentration of 100 μM [18]. According to the results, G1 and G2 Ds were non-toxic at most concentrations for all three cell lines, while they were slightly toxic at 100 μM. The G3 D was tolerated only at concentrations <1 μM by RAW 264.7 and U87 cells and concentrations « to 5 μM by HDF cells. The G4 and G5 Ds were toxic at very low concentrations (0.1 μM). Cationic Ds were also functionalized via carbamate coupling with monobenzylated tetraethylene glycol (BnTEG) and via amidation with the same species modified with a carboxylic acid linker (BnTEGCOOH), obtaining OH-Ds. Neurotoxicity of the prepared Ds was also evaluated on SH-SY5Y neuroblastoma cells [18]. A comparison was made on the cytotoxic effects of TMP-G4-NH_3_^+^TFA^−^ and TMP-G4-OH, those one of a PAMAM-D (SS-G3-PAMAM-NH_3_^+^TFA^−^) and of the SS *bis*-MPA dendrimer, with an equal number of amino end groups (SS-G4-NH_3_^+^TFA^−^). Concerning G3 and G4 Ds, OH-Ds were, in general, less toxic than NH_3_^+^, which instead caused only a slight reduction in neurotoxicity, when compared to commercially available PAMAM Ds. Overall, all NH_3_^+^ Ds proved considerable cytotoxicity and with regard to their antimicrobial activity against *E. coli*, only the TMP-G3-NH_3_^+^TFA dendrimer, while being non-toxic toward human cells, presented excessively high MIC values (MIC = 203 μg/mL) [18].

A second study on dendrimer-like polyester-based molecules, prepared using bis-HMPA as a building block and peripherally functionalized with an amino acid (β-alanine) in order to confer the cationic character essential for exerting the antibacterial activity on contact, was recently reported by Chen et al. (2020) [96] (Table 8).

A combinatorial library of first- to third-generation (G1–G3) dendrons, which juxtapose a cluster of cationic charges with a hydrophobic alkyl chain of different length (from C2 to C14), were synthetized by the authors, using the so-called “molecular umbrella” design approach. Eighteen amphiphilic cationic dendrons (G1–G3/C2–C14) were obtained and their antimicrobial activity was first evaluated against *E. coli* and *S. aureus* bacterial species. Among the compounds of each generation, the most active were those bearing the longest C14 chain (Table 8), confirming that a greater hydrophobicity could help dendrons diffuse from OM toward the CM (in the case of Gram-negative bacteria) and/or through the CM destroying it and inactivating the bacterium. These compounds, tested for their hemolytic toxicity, showed that HC_50_ values increased with the generation of the devices (from 10 to 5000 μg/mL), thus affirming the absence of hemolytic toxicity for G3/C14 (HC_50_ = 5000 μg/mL), associated with a potent antimicrobial activity against both the bacteria species (MIC values = 7.8 and 3.9 μg/mL, respectively) [96]. The greater hemolytic activity of the lower-generation dendrons is probably due to their greater hydrophobicity, deriving from the presence of the same C14 chain, and a lower number of cationic charges, which helps such molecules diffuse through cells membranes, thus improving their non-specific lytic activity. The killing activity of the third-generation dendron G3/C14, considered to be the leading candidate as a novel antimicrobial agent, since it is endowed with high efficacy and no hemolytic toxicity, was evaluated against *E. coli* at a concentration of 2× MIC (16 μg/mL). The results showed that the number of viable *E. coli* cells decreased immediately, and that within the first 30 min, the number of *E. coli* viable cells was reduced by approximately 2-logs (99.7% killing). Furthermore, after 1 and 2 h of incubation, reductions of 3.8 and 4.5-logs (99.9985 and 99.9997% killing) respectively were observed [96]. Considerably, the rate of bactericidal action was more rapid than the rate at which these polyester bis-HMPA-based dendrons typically are degraded by hydrolysis (37 °C, pH 7) [18]. Further investigations into G3/C14 antimicrobial activity against a large panel of bacterial strains showed a potent broad-spectrum antibacterial potency against several pathogenic microorganisms, including MRSA and *A. baumannii*. Investigations into the mechanism of action on representative examples of dendrons from the library developed by Chen’s group, revealed that cationic dendrons indeed exerted a potent lytic activity against *E. coli* membranes, probably supported by the presence of the hydrophobic chain pendant from the dendron structure. Cytotoxicity essays on HeLa cell lines revealed that within the library, G2/C14 was cytotoxic (LC_50_ = 32 μg/mL), G3/C14 moderately cytotoxic (LC_50_ = 85 μg/mL), whereas G3/C2 was completely non-toxic, up to the highest concentration tested (LC_50_ > 250 μg/mL), but also completely inactive against the pathogen [96].

Finally, very recently, three fifth-generation polyester-based Ds (G5Ds), characterized by a biodegradable inner matrix and a surface decorated with amino acids, were selected among the library of 15 polycationic homo- and hetero-Ds [93] that probably inspired the uncharged scaffolds reported later by Stenström et al. [18]. Those selected were of fifth generation, contain *L*-lysine (G5K), *L*-histidine (G5H) or a 50/50 mixture of both amino acids (G5KH) and all of them possessed 192 peripheral cationic groups and the ability to create electrostatic interactions with phosphate anionic charges [93]. These Ds, formerly synthesized for gene therapy applications, unlike compounds previously studied as antimicrobial agents [18,96], possessed a polycationic character in the presence of natural *L*-amino acids and, besides the potent G3/C14 previously discussed and described as moderately cytotoxic, totally lack toxicity in human HeLa in vitro cell lines [93]. G5Ds were investigated as novel antimicrobials agents against different clinical bacterial strains, including multidrug-resistant Gram-positive and Gram-negative bacteria [1]. Interestingly, all G5Ds (MICs = 0.5–33.2 μM), and particularly G5K (0.5–2.1 μM), displayed remarkable activity against non-fermenting Gram-negative species such as *P. aeruginosa*, *S. maltophilia* and *A. baumannii*, irrespective of their antibiotic resistance (Table 8) [1]. G5K activity was comparable to that of the previously described 3G/C14 amphiphilic dendron against *P. aeruginosa* (MIC = 2.1 μM vs. 2.2 μM) and greater than the activity showed against *A. baumannii* (MIC = 1.1–2.1 μM vs. 4.4 μM). As can be observed, G5K proved to be more active than the potent colistin (2.1 vs. 3.19 μM) against *P. aeruginosa*, responsible for alarming healthcare-associated infections [1]. Moreover, in time-killing experiments and turbidimetric studies, G5K displayed an unexpected rapid non-lytic bactericidal activity on *P. aeruginosa*, probably due to the absence of strongly hydrophobic residues, such as alkyl chains of the cationic umbrella molecules developed by Chen et al. [96], which instead proved to possess lytic behavior. In this study, conducted by Schito and Alfei [1], *E. coli*, along with *P. mirabilis*, *K. pneumoniae* and several Gram-positive species were considered refractory to G5Ds [1], since the MIC values obtained by the devices were >32–33 μM, a much higher value than that of the different antibiotics active today. On the contrary, the second generation of polyester-based alanine-modified cationic Ds reported by Stenström et al. [18], and similar to G5Ds, was considered active against *E. coli* even though it showed a MIC value of 100 μM.

### 4.5. Ammonium-Terminated Dendrimers

As happens among cationic antimicrobial polymers [2], also among CADs a class is represented by dendrimers which, despite having structurally different internal matrices, possess peripheral protonated amino groups, responsible for their cationic character and essential for their antibacterial potency. Table 9 summarizes the most representative of antimicrobial ammonium-terminated Ds and some of their variations developed in the last decade.

Second-generation (G2) ammonium-terminated Ds were prepared by synthesizing Boc-protected precursor Ds (G2-Boc), with a divergent synthetic strategy through polymerization of t-butyloxycarbonyl-aminoethyl acrylate (Boc-AEA), followed by deprotection treatment with trifluoroacetic acid (TFA) to remove Boc groups [97]. In plate killing essays, the G2 dendrimer obtained showed high antimicrobial activity against *E. coli* and *S. aureus* species (MBC_99.9_ = 3–8 μg/mL and 4 μg/mL respectively), associated with very low hemolytic toxicity (<10% at concentration of 1024 μg/mL). Interestingly, since net cationic membrane-disruptive antimicrobials may elicit an immune attack when administered intravenously, Xu’s same group [97], having cloaked a G2 dendrimer with poly(caprolactone-b-ethylene glycol) (PCL-b-PEG), obtained a new antimicrobial, G2-g-(PCL-b-PEG), which exhibits neutral surface charge. The neutral dendrimer was able to kill >99.9% of the inoculated bacterial cells, at ≤8 μg/mL, it exhibited good colloidal stability in the presence of serum and it showed insignificant hemolytic toxicity, even at concentrations ≥2048 μg/mL. The authors attributed the maintained antimicrobial activity of the neutral dendrimer to the degradation of the neutral shell due to bacterial lipase and the consequent exposure of the membrane-disruptive bactericidal cationic G2 *core*.

Semisynthetic ammonium-terminated gallic acid-triethylene glycol Ds (GATG) developed by Leire et al. (2016) have been shown to be able to interact with bacteria, thus attracting researchers as potential multivalent macromolecules for the development of new antimicrobials [98]. GATG Ds, due to the presence of primary amines in their periphery, were capable of sequestering bacteria cells and to induce the formation of clusters in *Vibrio harveyi*, an opportunistic marine luminescent pathogen. This property was dependent on dendrimer generations and was more potent than that of poly(*N*-[3-(dimethylamino)propyl]methacrylamide) [p(DMAPMAm)], a cationic linear polymer previously shown to cluster bacteria. The authors also investigated the bacteria viability within the formed clusters and the quorum sensing activity, responsible for light production in *V. harveyi*. The results suggested that GATG Ds may activate microbial responses by maintaining a high concentration of quorum sensing signals within the clusters and by increasing permeability of the microbial OM, with the growth of their generation. In particular, when bacteria cells were exposed to third-generation GATG Ds, more than 85% of population had OM damaged [98]. The GATG Ds developed by Leire et al. might constitute a valuable platform for the development of novel antimicrobial materials which can affect microbial viability and/or virulence.

By using the robust copper (I) catalyzed alkyne-azide cycloaddition (CuAAC) “click reaction” for dendrimer synthesis and for post-synthesis functionalization, Ladd et al. reported a versatile divergent methodology to construct ammonium-terminated Ds from the tetrafunctional *core* pentaerythritol [99]. Ds of generations 0–3 (G0–G3) were achieved with from 4 to 32 acetylene surface groups, which in G0–G2 Ds were subsequently used to covalently link cationic amino groups. The authors explored the bactericidal efficacy of the cationic amine terminated Ds G0-NH_3_^+^-G3NH_3_^+^ determining MIC and MBC values against the *E. coli* ATCC 11229TM strain [99]. The results asserted that the bacteriostatic and bactericidal activity of the prepared Ds depended on their size, functional end groups and hydrophilicity, as already established in many studies in the field. In particular, G0 dendrimer displayed poor antibacterial activity, with MIC values in the range 29–59 μM and MBC values > 59 μM, while G2 and G1 Ds displayed from good to strong antibacterial activity (Table 9).

In the authors’ opinion, the octacationic first-generation dendrimer was the most potent antimicrobial dendrimer displaying MIC values (0.9 μg/mL, 0.26 μM) and MBC values (4–8 μg/mL, 1.1–2.3 μM) that were lower than those of several previously studied Ds (Table 9) [99]. In our opinion, and considering when the MIC values are expressed in μM (which takes into account the MW of the Ds and which provides the actual equivalents of Ds able to exert a certain effect), rather than in μg/mL, the G2 dendrimer can be considered the most active antimicrobial device (MIC value = 0.12 μM vs. 0.26 μM). Furthermore, although the Ds developed by Ladd and colleagues may appear very promising, the absence of investigations into their cytotoxicity and hemolytic toxicity prevents them from being considered suitable for clinical applications.

#### 4.5.1. Ammonium-Terminated Phosphorous Dendrimers

By performing a sol-gel process, the fruitful association of a fourth-generation phosphorus dendrimer, peripherally modified with quaternized ethylene diamine and titanium dioxide (TiO_2_), was realized by Milowska et al. (2015) together with other three ones [59]. The hybrid mesoporous dendrimer-coated TiO_2_ obtained showed to be nanosized, in a crystalline and to possess a surface mimicking that one of PEI-Ds. It was investigated for its toxicity on red blood cells (RBCs), its cytotoxicity toward B14 Chinese fibroblasts and its antimicrobial activity against some strains of bacteria and yeast. The dendrimer proved appreciable antibacterial activity on *S. aureus* and *S. epidermidis* MSSE strains at MIC = 500 μg/mL in both cases. Nevertheless, considering that in hemolytic and cytotoxicity essays it showed a HC_50_ value of 100 μg/mL and an IC_50_ of 25 μg/mL, its antimicrobial activity must be considered questionable.

#### 4.5.2. Ammonium-Terminated Carbosilane Dendrimers

Cationic carbosilane Ds have shown their potential to be used as devices in several biomedical applications including as bactericides [59]. Both silicon containing hyperbranched polymers and Ds are interesting as organic-inorganic hybrid materials, due to the presence of C–Si bonds, that confer high stability and very low hydrophilicity, favorable to the processes of bio permeability [60]. Nevertheless, when preparing cationic materials for antimicrobial uses, hydrophobicity must be tuned by functionalizing the macromolecules with cationic groups, thus achieving good levels of water solubility, a pivotal requirement for antibacterial studies [60].

In this context, among other cationic materials, including hyperbranched polymers, some second- and third-generation (G2 and G3) cationic Ds, based on a carbosilane scaffold, but peripherally decorated with ammonium groups, were prepared by Ortega et al. (2011) and proposed as antibacterial agents [60]. Although slightly less active that the hyperbranched analogous, the cationic carbosilane Ds exhibited considerable antimicrobial activities against the Gram-negative *E. coli* and against the Gram-positive *S. aureus*, with MIC values in the range of 16–64 μg/mL and 4–8 μg/mL, respectively. MBC values, also calculated against the same strains, were in the ranges of 32–54 μg/mL and 8 μg/mL, respectively. The materials studied by Ortega and colleagues [60] seem very promising as novel broad-spectrum antimicrobial agents, but, since research on their cytotoxicity, hemolytic toxicity, and selectivity is missing, their suitability for clinical uses cannot be taken for granted.

Recently, Fernandez and colleagues (2019) synthetized G1 and G2 carbosilane dendrons and, using a selection of three different CAMPs, created a series of nanocomposites either by covalent links or by physical non-covalent interactions, with the aim of establishing synergistic or collaborative relations between both types of molecules [100]. The authors speculated that the carbosilane dendrimer structures, already known for their on contact antibacterial potency against Gram-positive and Gram-negative bacteria strains and for their poor tendency to induce resistances probably due to their easier penetration into the phospholipid bilayer could help to protect or transport AMPs inside bacterial membranes [100]. The antibacterial activity of ammonium-terminated carbosilane Ds (namely 3 and 4) was assessed against *E. coli* and *S. aureus* strains and was compared to that of the four nanocomposites [namely 11, 14 (covalent nanoconjugates) and 15, 16 (non-covalent nanoconjugates)] obtained. The MIC values calculated for both groups of compounds were comparable in the case of *E. coli*, but not in the case of *S. aureus*, since the nanocomposites (11, 15) displayed a significantly higher antibacterial activity than that of corresponding dendrons (3) (MIC values of 32 and 64 μg/mL vs. 128 μg/mL, respectively). This trend also applied to the nanocomposite (16), when compared to the corresponding dendron (4) (MIC value of 4 μg/mL vs. 8 μg/mL) (Table 9) [100].

#### 4.5.3. Ammonium-Terminated Ruthenium and Zinc Encased Phthalocyanines (Pcs) Dendrimers

An appealing approach to fight superficial and localized infectious diseases caused by MDR bacteria includes the antimicrobial photodynamic therapy (aPDT), which consists of an adequate combination of three factors: light, oxygen and a light-active compound defined photosensitizer (PS). Their combination leads to the formation of highly reactive oxygen species (ROS), responsible for the cytotoxic damage. Since Pc derivatives including metal *core* atoms such as zinc, ruthenium, or silicon have photosensitizing activity, they hold promise for various biomedical applications, including cancer therapy. However, due to their highly hydrophobic nature, Pcs require appropriate functionalization or combination with delivery systems to allow adequate administration.

In this regard, to obtain photosensitizers macromolecules capable of photoinactivation performance against representative microorganisms, the Gonzalez group (2017) synthetized two families of ammonium-terminated phthalocyanine Ds [101]. Four different categories of photosensitizer cationic Ds, made of zinc and ruthenium Pcs encased in multi-cationic Ds (namely ZnPc, ZnPc1, RuPc and ZnPc1) were essayed against *S. aureus*, *E. coli* and *Candida albicans* strains, upon red light irradiation, to assess their potential use as broad-spectrum photo-inactivating agents. All the Pcs Ds were more active against *S. aureus* than against *E. coli*, and zinc-coordinated Pcs were more active than ruthenium Pcs against both bacteria species. Octacationic ZnPc1 was slightly more powerful than tetracationic ZnPc against *S. aureus*, while the contrary was observed against *E. coli.* Vice versa, tetracationic RuPc was more active than octacationic RuPc1 against *S. aureus*, since RuPc1 was practically ineffective against *E. coli*. In fact, it caused a minimal reduction of the colony forming units per milliliter (CFU/mL), at the maximum tested concentration of 10 μM in the presence of the higher light-doses of 60 J cm^−2^ [101]. In particular, when tested against *S. aureus*, the most active compounds ZnPc1 and ZnPc produced over 6-log10 CFU/mL reductions at 1 μM in the presence of the higher light-doses and a bactericidal effect (3-log10 CFU/mL reduction) when irradiating at 60 J cm^−2^ with a concentration of 0.6 μM (Table 9). When essayed against *E. coli*, reductions over 6-log10 CFU/mL were achieved at concentrations as low as 2.5 μM for ZnPc and 5 μM for ZnPc1 at 30 J cm^−2^ light dose (Table 9) [101].

### 4.6. Cationic Organometallic Ds

Organometallic Ds, are a class of saline macromolecules, turned cationic due to the presence of a metal, with a plethora of applications. In this regard, a new class of organometallic Ds, containing iron and different types of counteranions, with tunable activity against MDR Gram-positive bacteria including MRSA and *Enterococcus faecium* (VRE), were synthetized by Abd-El-Aziz et al. (2015) [102]. Interestingly, these Ds were non-cytotoxic to human epidermal keratinocytes cells (HEka), to human foreskin BJ fibroblast cells, and to human breast adenocarcinoma cells (HTB-26). Furthermore, they were not hemolytic for mammalian red blood cells, even at the highest tested concentration of 128 μg/mL. For these reasons, such Ds can be considered to be potential antimicrobial platforms for topical applications [102].

The cationic organometallic Ds prepared by Abd-El-Aziz, are both cationic and redox-active macromolecules and therefore can exert a dual-mode antibacterial activity, by a disruptive action on the microbial membrane and by an oxidative effect causing oxidative stress on bacteria. Actually, nine Ds were prepared (DEN1-DEN9) and their antimicrobial activity was essayed against a broad spectrum of infection-causing pathogens, including Gram-positive and Gram-negative bacteria, i.e., *S. aureus* ATCC 33591 (MRSA), *S. warneri* ATCC 17917, *E. faecium* EF379 (VRE), *P. aeruginosa* ATCC 14210, *P. vulgaris* ATCC 12454, and mycetes as *Candida albicans* ATCC 14035. At the tested concentrations (<128 μg/mL), all Ds were inactive against the Gram-negative bacteria and *Candida albicans*, whereas all Ds, except for DEN 9, showed MIC values in the ranges of 1.8–15 μM, 2.2–22 μM, 2.1–12 μM against MRSA, VRE, and *S. warneri* respectively. DEN 2 was the most active dendrimer against MRSA and VRE strains (MIC values of 1.8 and 2.2 μM, respectively) with an IC_50_ value against HTB-26 of 20 μM, while DEN7, DEN8 were the most active against *S. warneri* (MIC values of 2.2 and 2.1 μM, respectively) and did not present cytotoxic activity on HTB-26 cells [102].

## 5. Conclusions and Considerations of the Authors

As can be observed in Figure 4, the antibacterial activity of the cationic Ds achieved through a nanobiotechnology approach in the last decade has been evaluated on 30 bacterial species. *P. aeruginosa*, *E. coli* and *S. aureus* were the most studied bacterial species, followed by *A. baumannii*, *K. pneumoniae*, *B. subtilis*, *S. epidermidis* and other minor species (Figure 4) [1,16,17,18,43,44,50,52,53,55,56,58,59,60,62,63,64,65,66,67,68,69,70,71,72,73,74,75,76,77,78,79,80,81,82,83,84,85,86,87,88,89,90,91,96,97,98,99,100,101,102].

As for *P. aeruginosa*, the most studied bacterial species, according to available data expressed in μM, among the structural categories of antibacterial cationic dendrimers developed, the compound with the strongest activity was a PAMAM-D, followed by a peptide dendrimer (PD), followed by an amino acid-modified polyester-based dendrimer (PEAAM-D) and, finally, by a PEI-D, which showed a MIC value higher than that of PAMAM-D, PD and PEAAM-D of 405, 18, and 3.9 times respectively (Figure 5) [1,50,55,80].

In general, the scenario obtained by examining the progress made in the development of new CADs and, in particular, of cationic ones, highlights that nanobiotechnology offers the possibility to engineer different structures of promising Ds, biomimetics of known cationic antimicrobials, which act as membrane disrupters, usually with better performance, mainly due to their multivalence. However, despite the very promising results deriving from the evaluations of the different CADs, developed and tested on representatives of Gram-positive and Gram-negative bacteria, since they were born and raised only in the last decade, these devices require further studies and more rationalized investigations. In this regard, analyzing the structural classes of Ds mainly studied, the investigations carried out, the strains of bacteria taken into consideration (Figure 3), the reported antibacterial activities, and the presentation of the results, major discrepancies and imbalances occur. Although many dendrimer structures, *per se* cationic or cationic yieldable through post-synthesis functionalization have been developed in the last 30 years, only a few categories have been taken into consideration as antimicrobial agents in the last ten years. Moreover, noticeably in Figure 6, while there are dendrimer structures that have been extensively investigated by modifying native structures with several different approaches, by researching for possible synergistic co-operations with conventional antibiotics, by searching for different types of bioactive residues, and by studying structure-activity relationships etc., other Ds have been considered only marginally, despite the first promising results.

Although generally, PAMAM Ds and PPI Ds were and are the most extensively studied devices for applications in multiple sectors, including nanomedicine, until now peptide dendrimers are the cationic dendrimer structures investigated more as antibacterial devices, perhaps due to their close relationship with natural and synthetic CAMPs, whose activity has been improved due to the multivalence allowed by the generational structure of the Ds. On the contrary, organometallic cationic Ds (OMC-Ds) received very poor consideration and, to the best of our knowledge, only one study reporting their antimicrobial activity has been found. Considering that, even if ineffective against Gram-negative bacteria, they proved considerable activity against Gram-positive species, this class of CADs deserve further investigation in the future. Amino acid-modified polyester-based Ds, extensively studied for drug delivery, gene therapy, or as solubilizing devices, have only very recently attracted researchers’ interest as antibacterial agents, with very appealing results at least in two cases out of three. These types of Ds, by merging an uncharged ester-type inner matrix with the presence of amino acids as cationic moieties, could mimic peptide Ds and CAMPs. In addition, due to the high generations easily achievable, they could provide very high multivalence and density of positive charges, helpful for exerting high antibacterial activity on contact, maintaining low levels of toxicity, due to their high biodegradability. For these reasons, more in-depth investigations, concerning hydrophilic or amphiphilic Ds based on polyester structures modified with amino acids, are suggested.

Regarding the study of bioactivity of the developed Ds, in order to allow a critical comparison, the types of investigations to be performed should be standardized, e.g., by following the typical microbiologic path that applies during the preliminary evaluation of the antibacterial potency of a new substance. This path should include the determination of the minimum inhibitory concentration of the growth of bacteria (MIC), then the determination of the concentration capable of killing the bacterium (MBC) confirmed by the set-up of the time-killing experiments, which monitor the logarithmic reduction of bacterial inoculos exposed to MBC values of the test compound over 24 h. By reviewing the studies performed, and as can be observed in Figure 7, the MIC values have been reported in the 43% of cases, the MBC ones in 24%, and the time-killing experiments were performed only in 6% of the case studies; and in one case only they were representative of a 24 h experiment. Histogram plots have frequently been reported, showing the reduction in bacteria cell viability after exposure to different concentrations of Ds and, sometimes, without a known antibiotic as a reference.

Furthermore, considering the harmful infections caused by drug resistant bacteria, also supported by the formation of the biofilm, further experiments are suggested to evaluate the potency of the developed Ds in inhibiting biofilm formation (MBIC), killing bacterial cells under biofilm-forming conditions (MBEC) and dispersing established detrimental biofilm. Furthermore, investigations concerning both the assessment of hemolytic toxicity, a major concern of cationic devices acting on the bacterial membrane, and the cytotoxicity of CADs to eukaryotic cells should be improved. Hemolytic toxicity and cytotoxicity were determined in only 38% and 19% of cases respectively, while they should always be determined to judge the real clinical applicability of the synthesized materials.

Finally, to allow the scientific community interested in the field to easily compare the antibacterial activity of developed Ds, the way results are presented (MIC, MBC values) should also be standardized. In this regard, a debate could be opened, relating to the expression of MIC/MBC values as μg/mL, usually adopted by microbiologists, or as μM, perhaps preferred by chemists dealing with high MW macromolecules. Although as reported earlier in the main text of this review, the MIC values expressed in μM, which considers the MW of the Ds and which provides the actual equivalents of D able to exert some effect, could indeed provide comparable values, a suggestion to resolve the debate could be to express the results in both ways.

## Figures and Tables

**Figure 1 nanomaterials-10-02022-f001:**
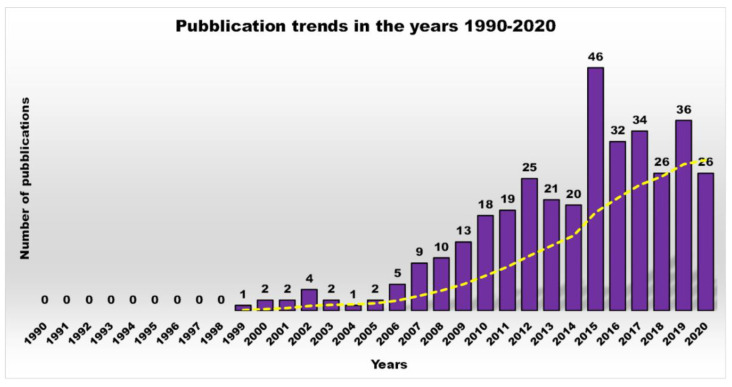
Number of publications as a function of time, obtained by typing the key words “antimicrobial dendrimers” in Scopus.

**Figure 2 nanomaterials-10-02022-f002:**
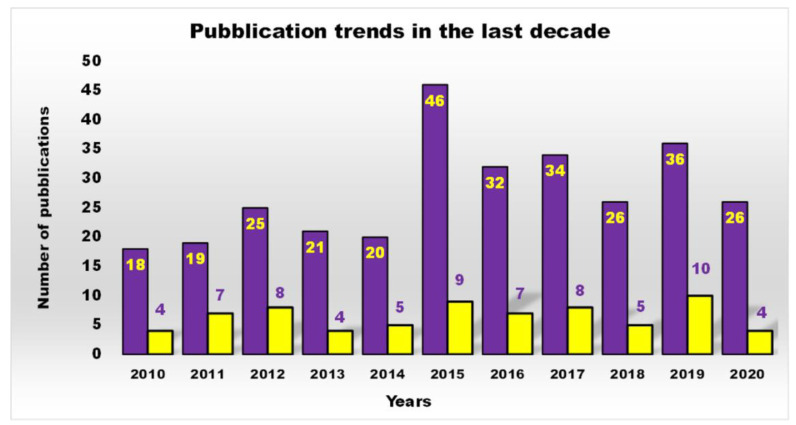
Number of publications, as a function of time, obtained by typing the key phrase “antimicrobial dendrimers” in Scopus (purple bars) and that obtained by typing “cationic antibacterial dendrimers” (yellow bars), topic of the review.

**Figure 3 nanomaterials-10-02022-f003:**
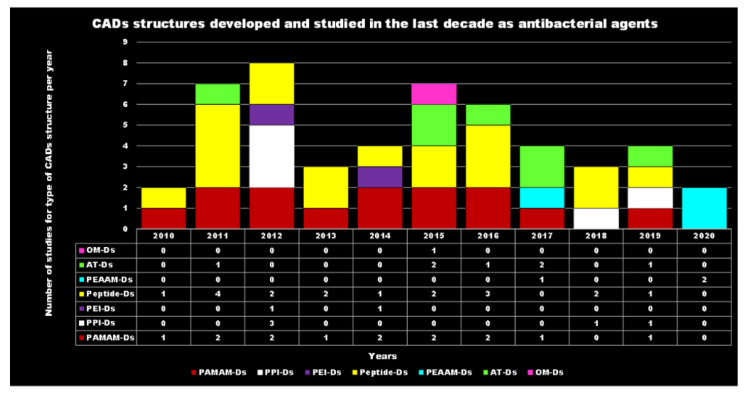
CADs developed in recent decades.

**Figure 4 nanomaterials-10-02022-f004:**
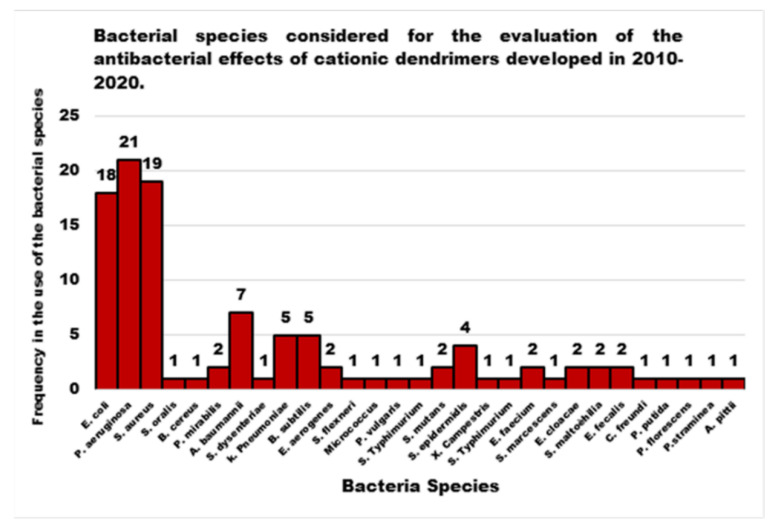
Bacterial species employed to evaluate the antibacterial activity of cationic Ds.

**Figure 5 nanomaterials-10-02022-f005:**
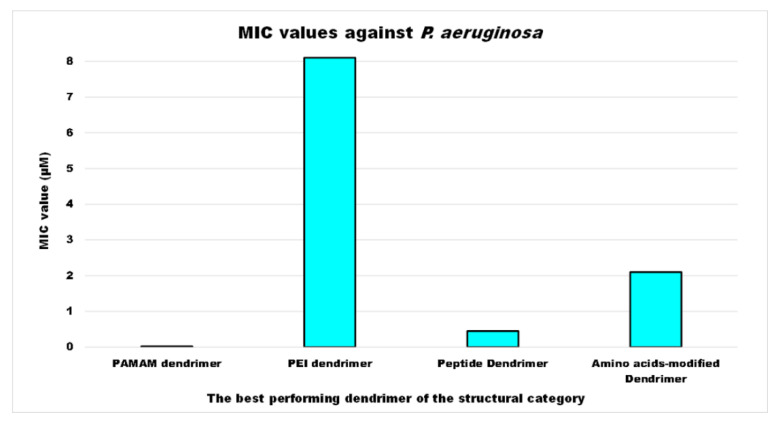
Antibacterial activity of the best performing compounds for one structural category of CADs against *P. areruginosa*.

**Figure 6 nanomaterials-10-02022-f006:**
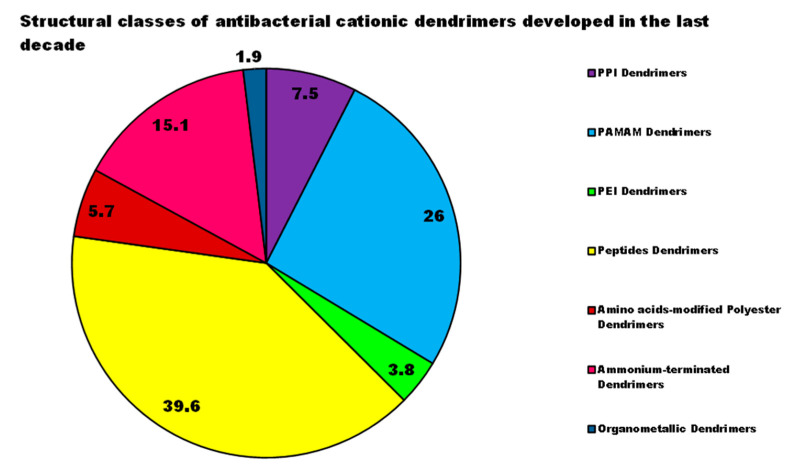
Percentage of the main classes of CADs developed in the years 2010–2020.

**Figure 7 nanomaterials-10-02022-f007:**
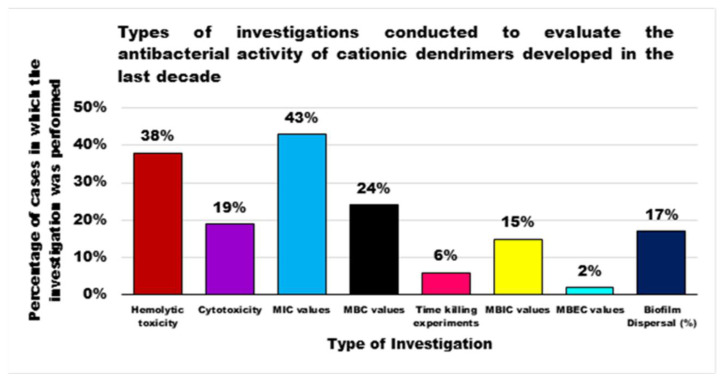
Percentages of the main biological investigations conducted on CADs developed in the years 2010–2020.

**Table 1 nanomaterials-10-02022-t001:** Bacteria biofilm contribution in the antibiotic’s inactivation and antibiotic treatment failure.

Reasons for Failure of Antibiotics	Biofilm Function	Factors	Bacteria	Inactivated Antibiotics	Ref.
Hampered antibiotic penetration	Anti-spread barrier	EPS	*P. aeruginosa* (exopolysaccharides)	Cationic antibiotics aminoglycosides	[26,27]
Presence of antibiotic-degrading enzymes	To provide β-lactamases (β-LS)	↑ β-LS	*K. pneumoniae*	Ampicillin	[28]
			*P. aeruginosa*	Imipenem Ceftazidime	[29]
Increased biofilm resistance	To provide eDNA	↑ eDNA ↓ Mg^2+^	*P. aeruginosa* (Spermidine) *Salmonella enterica*	Cationic Peptides Aminoglycosides	[30,31,32]
Presence of persistent cells	To cause gradients in nutrients and oxygen concentration To promote differentiation in cell growth	Endogenous stress TA ^1^-systems	*P. aeruginosa* *E. coli*	Rifampicin Aminoglycosides	[33]
Presence of dormant cells		↓ Functions ↓ Energy ↓ Biosynthesis	*E. coli*	Fluoroquinolones	[34]
↑ Resistance to stress	To cause adaptive stress responses by heterogeneity	Changes in component/processes target of antibiotics	*P. aeruginosa*	Ofloxacin Gentamicin Meropenem Colistin	[35]
			*E. coli* K-12	Ofloxacin	[36]
↑ Exporting membrane proteins	To up-regulate the production of some efflux pumps	↑ Efflux pumps QS	*E. coli* *Enterobacter aerogenes* *K. pneumonia*	Multi-drugs	[37]
			*P. aeruginosa*	Azithromycin	[38]
Genetic diversity	To act as reservoir of genetic diversity by promoting plasmids transfer	Horizontal gene transfer (HGT) eDNA QS	*P. aeruginosa*	Aminoglycosides	[39]

^1^ TA = toxin/antitoxin; ↑ = improved, higher, increased; ↓ = reduced, decreased.

**Table 2 nanomaterials-10-02022-t002:** Main types of dendrimer structures developed in the past 40 years.

Dendrimer Structure	Name	Structural Features
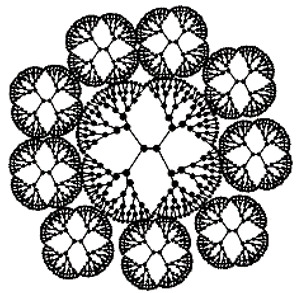	Tecto Ds	Central dendrimer with multiple peripheral Ds able of differentiate activities
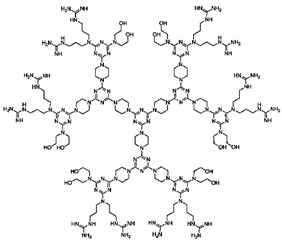	Triazine Ds	Triazine *core* with repeated triazine units
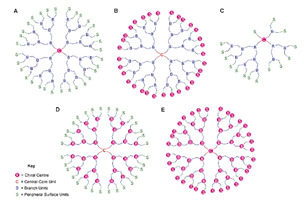	Chiral Ds	Stereogenic centers within the dendritic structure
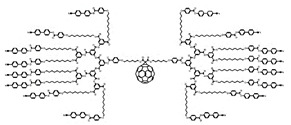	Liquid crystalline Ds	Block molecules with dendritic *core* and mesogenic terminal groups
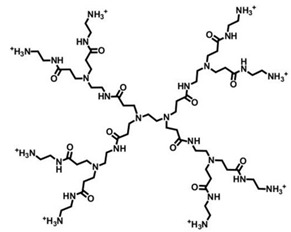	PAMAM Ds	Amidoamine *core* and amidoamine repeated units
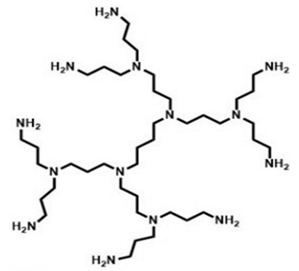	PPI Ds	Propylene diamine *core* and similar repeated units
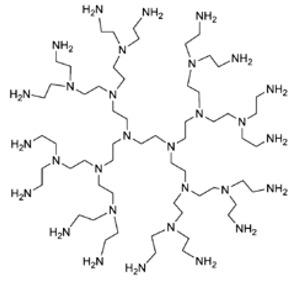	Poly(ethyleneimine) (PEI-Ds)	Ethylene diamine *core* and similar repeated units
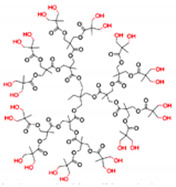	Polyester-based Ds	AB2 monomer repeated units linked by ester-type hydrolysable bonds
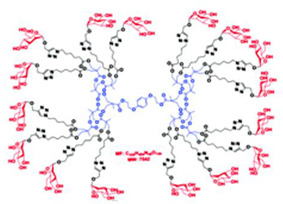	Glyco Ds	Ds containing saccharide units
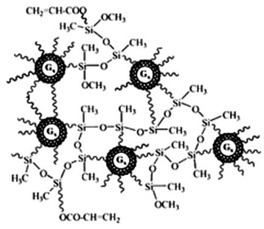	Polyamidoamine-organosilicon (PAMAMOS) Ds	PAMAM Ds containing silicon organic groups
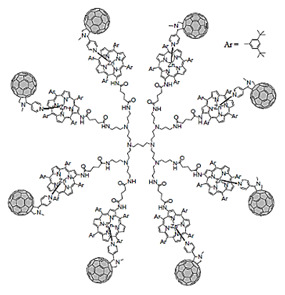	Fulleropyrrolidine Ds	Ds containing fullero pyrrolidine units
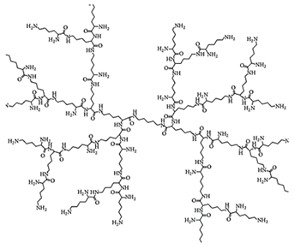	Poly(lysine) Ds	Ds containing *N*-lysine repeated units
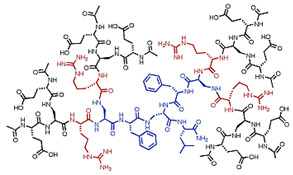	Peptide Ds	Dendrimer containing amino acids combined by peptide bonds
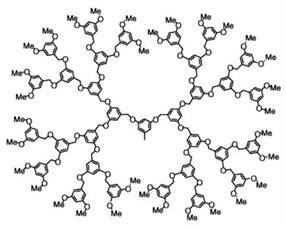	Polyether Ds	Ds with repeated units linked by ether-type bonds
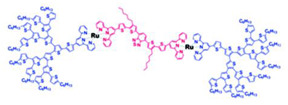	Metal Ds	Dendrimer with incorporated metal atoms
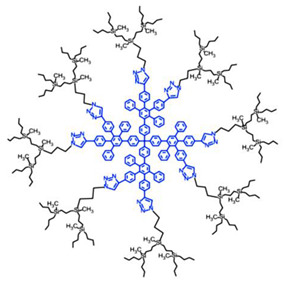	Hybrid Ds	Ds encompassing more than one type of dendrimer structure
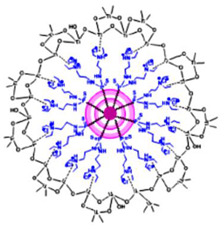	Ammonium-terminated phosphorous hybrid Ds	Hexachlorocyclo triphosphazene *core* with hydroxybenzyl phosphor hydrazone repeated units
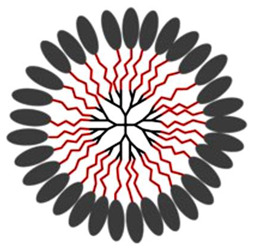	Mesogenic Ds	Ds containing parts responsible for the formation of a mesophase and liquid crystals
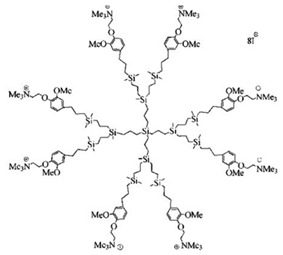	Carbosilane Ds	Ds containing silicon atoms

**Table 3 nanomaterials-10-02022-t003:** Main types of antibacterial Ds developed in the years 2000–2010.

Dendrimer Class	Mechanism of Action Proposed	Structure Characteristics	Advantages	Target Bacterial Species Toxins	Range of Activity ^2,3,4^
Glyco Ds	Interference with adhesion of toxins to eukaryotic cells (ECs)	1/2 G ^1^ PPI Ds 1 G ^1^ PAMAM Ds	↑ Interactions with carbohydrate-targets due to dendrimer multivalence	*Vibrio cholerae* toxin B *E. coli* enterotoxin	↑ 1000-times ^2,3^
					↓ adherence ^2,4^
		3 G 3,5-di-2-PAE(BA) ^5^		*Vibrio cholerae* toxin B	↑ 380 × 10^3^-times ^2,3^
	Interference with adhesion of bacteria to ECs	8-valent galabiose 1 G ^1^ PAMAM Ds 3,5-di-2-PAE(BA) ^5^		*Streptococcus suis*	MIC * 0.3 nM
	Interference with adhesion of bacteria to human erythrocytes	12-valent mannose 3G ^1^ PAMAM Ds		Type I fimbriated *E. coli*	MIC * 19 μM ↑ 400-times ^2,4^
	Interference with adhesion of bacteria to ECs	4-valent α-C-fucosyl-Ds	↑ Interactions with carbohydrate−targets due to dendrimer multivalence No cytotoxicity ↑ Stability	*P. aeruginosa*	IC_50_ ^§^ 0.14 μM
				Biofilm *(P. aeruginosa)*	IC_50_ ^§^ 10 μM
Cationic Ds	Electrostatic interactions Bacterial membranes disruption	Quaternized PPI Ds	↑ Positive charge density due to dendrimer multivalence	Recombinant *E. coli* (TV 1048)	EC_50_ ^#^ 0.14 μM ↑ 100-times ^2,3^
		1/2 G ^1^ ammonium-carbosilane Ds		*Staphylococcus aureus*	MIC * 8 mg/L (2G) 1 mg/mL (1G)
				*E. coli*	MIC * 64 mg/L (2G) 4 mg/L (1G)
		1-3 G ^1^ ammonium *N,N*-dimethyl-*N′*-allyl *N′*-ethylethylene diamine hydride-terminated carbosilane Ds	↑ Positive charge density due to dendrimer multivalence Water-soluble	*E. coli*	MBC ^$^ 1.65 mg/L (3G) 1.70 mg/L (2G) 3.65 mg/L (1G)
				*S. aureus*	MBC ^$^ 0.82 mg/L (3G) 0.85 mg/L (2G) 1.82 mg/L (1G)
		4G ^1^ nadifloxacin-loaded PAMAM Ds	↑ Positive charge density due to dendrimer multivalence Antibiotics solubilization by encapsulation	*E. coli*	As nadifloxacin
		4G ^1^ prulifloxacin-loaded PAMAM Ds			↑ 2-times prulifloxacin
		PEG/not PEG 5G ^1^ PAMAM Ds coatings	↑ Positive charge density due to dendrimer multivalence ↓ Cytotoxicity Prevention of implant-associated infections	*P. aeruginosa* (ATCC 19660)	EC_50_ ^#^ 1.5 μg/mL
				*P. aeruginosa* (clinical)	EC_50_ ^#^ 0.9 μg/mL
		6% PEG 3G ^1^ PAMAM Ds coatings	↑ Positive charge density due to dendrimer multivalence ↓↓↓ Cytotoxicity	*P. aeruginosa*	MIC * 25 μg/mL
		Hydroxyl-terminated 4G ^1^ PAMAM Ds	↑ Positive charge density due to dendrimer multivalence	*E. coli*	Inhibition of bacteria growth and ascending in uterus ^6^
		Amoxicillin-loaded cross-linked 4G ^1^ PEG-PAMAM Ds	↑ Positive charge density due to dendrimer multivalence Injectable hydrogels ↑ 72 h residence time No cytotoxicity 240 h sustained drug release ^7^		
		Triclosan-loaded 4G ^1^ PPO ^8^/PAMAM/PAA ^9^ layer-by-layer device	↑ Positive charge density due to dendrimer multivalence ↑ Drug loading than surfactants 20 days sustained drug release ^7^	*S. aureus*	Inhibition of bacteria growth
		PPO ^8^ triamine *core* PAMAM Ds	Positive charge density due to dendrimer multivalence	*E. coli* *S. aureus* *K. pneumoniae* *B. cereus* *M. lutens* *P. vulgaris* *M. Smegmatis* *L. monocytogenes* *P. aeruginosa*	MIC * (μg/mL) 3.1–12.5 1.6–6.3 6.1–12.5 6.3–25.0 1.6–12.5 1.6–6.3 12.5 6.3–12.5 6.3–25.0
Anionic Ds	Imitating detergent activity	Amphiphilic Ds	↑ Negative charge density due to dendrimer multivalence Significant selectivity	*Bacillus subtilis*	EC_50_ ^#^ 41 μM
		PPO ^8^ Triamine/anionic PAMAM Ds	↑ Negative charge density due to dendrimer multivalence Significant selectivity	*E. coli* *S. aureus* *K. pneumoniae* *B. cereus* *M. luteus* *P. vulgaris* *M. Smegmatis* *L. monocytogenes* *P. aeruginosa*	MIC * (μg/mL) 12.5–25.0 12.5 12.5–25.0 12.5–25.0 6.3–12.5 6.3–12.5 12.5–25.0 6.3–12.5 12.5–25.0
Peptide-based Ds	Mimicking membrane-active antimicrobial peptides [2]	2, 4, 8-valent polylysine-Ds with Arg-Leu-Tyr-Arg or Arg-Leu-Tyr-Arg-Lys-Val-Tyr-Gly sequences on surface [D_2,4,8_(R4), D_2,4,8_(R8)]	↑ Positive charge density due to dendrimer multivalence Significant selectivity ↓ Hemolytic activity	*E. coli* *P. aeruginosa* *P. vulgaris* *K. oxytoca* *S. aureus* *M. luteus* *E. faecalis*	MIC * (μM) D_8_(R8)/D_8_(R4) 0.3/0.5–0.7 0.5/0.3–0.9 0.5/0.8–1.3 0.5/0.4–0.8 0.4/0.5–0.6 0.4/0.5–0.7 0.4/0.8–1.3
		4-valent Polylysine-D *core* with Gln-Lys-Lys-Ile-Arg-Val-Arg-Leu-Ser-Ala sequences on surface	↑ Positive charge density due to dendrimer multivalence Significant selectivity ↓ Hemolytic activity ↓ Cytotoxicity Good stability in plasma	*E. coli* *K. pneumoniae* *K. oxytoca* *E. aerogenes* *E. doacae* *P. mirabilis* *A. baumannii* *Citrobacter* *freundii* *B. cepacia* *S. aureus* *P. aeruginosa*	MIC * (μg/mL) ^10^ 8 4–16 64 8 4 64→128 16–32 16 64 128 4–8
		4-valent Lysine-D *core* with Trp-Arg on the surface	↑ Positive charge density due to dendrimer multivalence Significant selectivity ↓ Hemolytic activity ↓ Cytotoxicity ↓ Resistance development Capable of synergistic action	*E. coli res.*	MIC_50_ * (μg/mL) 4.5 ↓ 33.5% planktonic ↓ 93.5% biofilm
				*S. aureus res.*	MIC_50_ * (μg/mL) 16.0

D(s) = dendrimer(s); ^1^ G = generation; ^2^ compared to non-dendrimer materials or carbohydrate residues; ^3^ experiments in wells; ^4^ experiments on cell lines; ^5^ PAE(BA) = poly-aminoethoxy(benzoic acid); ^6^ in vivo experiments; ^7^ in vitro experiments; ^8^ polypropylene oxide; ^9^ polyacrylic acid; ^10^ MIC values have not been reported when >128 μg/mL; * MIC/MIC_50_ = minimum inhibitory concentration/minimum inhibitory concentration of 50% of bacteria tested; ^§^ IC_50_ = half maximal inhibitory concentration; ^#^ EC_50_ = Half maximal effective concentration; ^$^ MBC = minimum bactericidal concentration; ↑ = increased, improved, higher; ↓ = reduced, decreased, smaller; ↓↓↓ = strongly reduced.

**Table 4 nanomaterials-10-02022-t004:** Two-increase of primary peripheral amine groups subsequent to the increase of one generation.

PAMAM/PPI Ds Generation	Number of NH_2_ Peripheral Groups
0	4
1	8
2	16
3	32
4	64
5	128
6	256
7	512

**Table 5 nanomaterials-10-02022-t005:** Most representative positively charged antimicrobial PPI- and PAMAM Ds developed in the years 2010–2020 and antibacterial activity of the most effective among the synthetized series.

Dendrimer Structure	Proposed Mechanism	HC_50_ (μg/mL)	Cytotoxicity IC_50_ (μg/mL)	Target Bacteria	Activity (μg/mL)
LVFX (0.1 μg/mL) co-administered Highly maltose-modified 3G PPIs (PPI-G3-DS-Mal) [16]	Electrostatic interactions OM/CM damage OM/CM disruption	N.D.	Cell viability% (100 μM) B14 80 HepG 100 N2a 100 BRL-3A 100	*E. coli*	MBC_80_ (μM) 10
C16-DABCO-loaded mannose-terminated 4G PAMAM Ds [43]	Electrostatic interactions OM/CM damage OM/CM disruption	MHC (μM) 0.6	A549 human lung carcinoma cells Observed at 1.1 μM	*S. aureus* *S. aureus* (biofilm) *B. cereus* *P. aeruginosa* *E. coli*	MIC (μM) 0.13 33.3 0.13 2.0 1.1
G7 PAMAM-D [50]	Electrostatic binding Non-specific OM/CM disruption	N.D.	Cells viability (%) 55 HCT116 57 NIH 3 T3	*P. aeruginosa* *E. coli* *A. baumannii* *S. dysenteriae* *K. pneumoniae* *P. mirabilis* *S. aureus* *B. subtilis*	MIC/MBC 4–8/128–256 4–8/128–256 4–8/128–256 1–2/64–128 4–8/128–256 1–2/64–128 4–8/128–256 2–4/64–128
Self-assembly poly(aryl ether)-PAMAM-based amphiphilic Ds [53]	OM/CM disruption assessed by fluorescent assays OM/CM depolarization	N.D.	IC_75_ 250 (48 h) ^2^	*E. coli* *S. aureus*	MIC 62 31
				*E. coli* *S. aureus*	MBC 125 31
CdS/Ag_2_S (QDs)-loaded PAMAM Ds/ MWCNTs [62]	OM/CM damage ^5^ CM disruption ^5^ DNA damage by CdS/Ag_2_S penetration	N.D.	N.D.	*E. coli* *P. aeruginosa* *S. aureus*	GR ^1^% CdS (20 μg/mL) 87.2 68.9 46.7
				*E. coli* *P. aeruginosa* *S. aureus*	GR ^1^% Ag_2_S (20 μg/mL) 97.8 78.5 55.7
G2-G5 PPI-PO-Ds G2-G4 PPI-PEG-Ds G2-G5 PPI-SO-Ds G3-G5 PPI-PO-NO-Ds G2-G5 PPI-PEG-NO-Ds G2-G5 PPI-SO-NO-Ds G2-G5 PPI-NH_3_^+^-Ds [64]	General OM/CM disruption action Additional for NO-Ds: Oxidative and nitrosative stresses Reactive NO byproducts [(N_2_O_3_), peroxynitrite (ONOO−)] Membrane destruction via Peroxynitrite-induced lipid peroxidation Protein S-nitrosation DNA deamination	Data in the text	N.D.	*P. aeruginosa*	MBC (μM) 5 ^2^, 1 ^3^ 5 ^4^, 0.5 ^3^
				*S. aureus*	MBC (μM) 2.5 ^5^, 0.5 ^3^ 0.5 ^4^, 0.5 ^3^
				*S. aureus* MRSA	5 ^2^, 2.5 ^3^ 0.5 ^4^, 0.25 ^3^
G1 PAMAM-ED-NO-D G1 PAMAM-PE3/7-NO-D G1 PAMAM-PE5/5-NO-D G1 PAMAM-PE7/3-NO-D G1 PAMAM-PO-NO-D G3 PAMAM-PE7/3-NO-D [65]	General OM/CM disruption action Oxidative and nitrosative stresses Reactive NO byproducts [(N_2_O_3_), peroxynitrite (ONOO−)] Membrane destruction via Peroxynitrite-induced lipid peroxidation Protein S-nitrosation DNA deamination	N.D.	Cells ^6^ viability at MBEC values 1st G ED 35% 1st G PE3/7 30% 1st G PE5/5 100% 1st G PE7/3 105% 1st G PO 55% 3rd G PE7/3 90%	*P. aeruginosa*	MBC 5
				*P. aeruginosa* Biofilm	MBIC 15
G1-G4 NO-releasing-alkyl(QA)-PAMAM Ds [66]	General OM/CM disruption action Oxidative and nitrosative stresses Reactive NO byproducts [(N_2_O_3_), peroxynitrite (ONOO−)] Membrane destruction via Peroxynitrite-induced lipid peroxidation Protein S-nitrosation DNA deamination	N.D.	Cells ^6^ at MBC values 80–110%	*P. aeruginosa*	MBC 10
				*S. aureus*	MBC 10
1G NO-releasing octyl- and dodecyl-modified PAMAM Ds [68]	Electrostatic dendrimer-bacteria interactions OM/CM damage Fast NO-release kinetics from proton-labile N-diazeniumdiolate NO donors	N.D.	MBC 1000–2000 ^7^ 15–20%	*S. mutans* *S. mutans* biofilm	MBC (pH = 6.4) 15 MBIC 1000

D(s) = dendrimer(s); N.D. = Not determined; ^1^ growth reduction; ^2^ G2PPI-SO; ^3^ NO-releasing dendrimer; ^4^ G5PPI-SO; ^5^ G5PPI-propylene oxide; ^6^ L929 mouse fibroblasts; ^7^ HGF-1 fibroblasts.

**Table 6 nanomaterials-10-02022-t006:** Most representative antimicrobial PEI-Ds developed in the years 2010–2020 and antibacterial activity of the most effective ones among the series synthetized.

Dendrimer Structure	Proposed Mechanism	HC_50_ (μg/mL)	Cytotoxicity IC_50_ (μg/mL)	Target Bacteria	Activity (μg/mL)
PEI-D 4[N[(Ts)(2-(methyl)-5-aryl-1,3,4 oxadiazole)]] [55]	OM/CM damage OM/CM disruption by PEI fraction Electron donating action of heterocycle groups	N.D.	N.D.	*B. subtilis* *S. aureus* *S. epidermidis* *E. coli* *X. campestris* *S. typhi* *P. aeruginosa*	MIC 12.5 12.4 12.5 12.5 12.5 5 12.5
Unmodified *b*-PEI-D [56]	OM/CM damage OM/CM disruption	>4000	27→4000 (1 h) ^1^ 7–2305 (24 h) ^1^	*E. coli* *S. aureus*	MIC 250 16

^1^ HEp-2 cells.

**Table 7 nanomaterials-10-02022-t007:** Most representative cationic antibacterial peptide Ds developed in the years 2010–2020 and antibacterial activity of the most effective ones among synthetized series.

Dendrimer Structure	Proposed Mechanism	HC_50_ (μg/mL)	Cytotoxicity IC_50_ (μg/mL)	Target Bacteria	Activity (μg/mL)
Cationic peptide dendrimer RW4D [58]	OM/CM damage OM/CM disruption	1962	N.D.	*E. coli* *A. baumannii* *S. aureus*	IC_50_ 3.9 15 42
G2-G4 peptide Ds [73]	OM/CM damage OM/CM disruption	μM <6.3 >200	N.D.	*B. subtilis* *S. aureus* *E. coli* *P. aeruginosa*	MIC (μM) 0.37 2.5 0.45 13.8
G3 peptide-D (3GKL) [74]	OM/CM damage OM/CM disruption	840	N.D.	*A. baumannii* *P. aeruginosa* *A. baumannii* *P. aeruginosa*	MIC_50/90_ 8/8 8/8 MBC_50/90_ 8/8 8/8
Amphiphilic peptide Ds with Ornitine *core* [75]	OM/CM damage OM/CM disruption	μM 100 (0–70%)		*S. aureus* ^1^ *S. aureus* ^2^ *E. coli* *P. aeruginosa*	MIC (μM) 0.93 0.46 1.85 7.8
Peptide Ds [76]	Electrostatic binding Insertion into lipid bilayer OM/CM damage OM/CM disruption	MHC 20→10,000		*B. subtilis* *E. coli* *P. aeruginosa* *S. aureus* ATCC 25923 *S. aureus* 887 *S. epidermidis* ATCC 14990 *S. epidermidis* J147 *En. faecium* ATCC 19434 *En. faecium* Van B E38-10 *E. coli* HB101 (PAT266) *E. coli* DC2	MIC (H1, bH1) ^#^ 2 3.9 18 MIC (H1, bH1) ^#^ 24 12 24 24 2 8 32 32
Amphiphilic dimeric peptide-D (SB056) [77]	Electrostatic binding Insertion into lipid bilayer Membranolysis by lipid-induced aggregation	159	N.D.	*A. baumannii* *E. cloacae* *E. coli* *K. pneumoniae* *P. aeruginosa* *S. maltophilia* *P. mirabilis* *S. marcescens* *E. faecalis* *E. faecium* *S. epidermidis* *S. aureus*	MIC 4 4 2 8 (MBC) 4 4 8 32 >128 32 8 8 (25.6) ^3^ 32 (51.6) ^3^ 64 (MBC)
Dimeric peptide-D (SB056-1) [78]	Electrostatic binding Insertion into lipid bilayer Membranolysis by lipid-induced aggregation	87	N.D.	*E. coli* *S. aureus* *E. coli* *S. aureus*	MIC 16 32 MBC 16 32
G2 KW peptide-D (2D-24) [14]	Electrostatic binding Insertion into lipid bilayer OM/CM damage OM/CM disruption Biofilm and alginate penetration	>1000	>50 ^4^	*P. aeruginosa* (PAO1) *P. aeruginosa* (PDO300)	MIC GR ^5^, 46.5 μg/mL 77.5 94% (biofilm) 77.5 94% (biofilm)
G3 peptide Ds [80]	Electrostatic binding Non-specific OM/CM damage OM/CM disruption	MHC G3KL 840 DG3KL 680	N.D.	*P. aeruginosa* PAO1 *E. coli* DH5α *B. subtilis* BR151 *P. aeruginosa* ^6^ *P. aeruginosa* ^7^ *A. baumannii* ATCC19606 *E. coli* W3110 *E. aerogenes* 13048	MIC G3KL/DG3KL ^#^ 2/4 4/1 3/2 4/4 4/4 8/16 4/2 64/32
G3 peptide-D (T7) [81]	OM/CM damage OM/CM disruption Leakage of cell contents	MHC >2000	N.D.	*P. aeruginosa* PAO1 *P. aeruginosa* ^6^ *E. coli bla* ^8^ *E. coli* ^8^ *C. freundii bla* ^8^ *E. cloacae bla* ^8^ *K. pneumoniae* ^8^ *K. pneumoniae bla* ^8^	MIC 4 8 8 8 4 8 16 16
G3 peptide-D (T7) [82]	OM/CM damage OM/CM disruption Leakage of cell contents	MHC 500	N.D.	*P. aeruginosa* PAO1 *P. aeruginosa* ^6^ *E. coli* ^8^ *K. pneumoniae* ^8^ *K. pneumoniae bla* ^8^ *S. aureus* ^8^	MIC 2 4 8 32 16 32
LecA and LecB ligands glycopeptide Ds (GalAG2, GalBG2, FD2) Heteroglycopeptide-Ds (Het1G2-Het8G2) β-fucoside-D (FucC6G2) Lewis^a^ analogous glycopeptide Ds (Le^a^xG2) Non-glycosilated peptide Ds (AcG2xK, NG2) [91]	Inactivation of LecA and/or LecB Electrostatic binding OM/CM damage OM/CM disruption	N.D.	N.D.	*P. aeruginosa* * Biofilm	MBC (μM) 13 (Het7G2) 13 AcG2xK) 2.5 NG2) MBIC (μM) 20 (FD2) 20 (GalAG2) 20 (GalBG2) 13 (Het7G2) 9 (FucC6G2) 30 (Le^a^xG2) 13 (AcG2xK) 2.5 (NG2)

N.D. = Not determined; K = *L*-lysine; H = *L*-histidine; ^1^ ATCC 25923; ^2^ ATCC 43300; ^3^ antibiofilm activity; ^#^ MIC values are referred to the specific dendrimers named H1, bH1, G3KL and DG3KL in the original reference work because they were found to be the most active of a vast library of compounds; ^4^ cytotoxicity to IB3-1 epithelial cells; ^5^ biofilm cells; ^6^ clinical isolates of *P. aeruginosa* (ZEM9A, ZEM1A, PEJ2.6, PEJ9.1) resistant to β-lactams (cephalosporins, carbapenems), aminoglycosides (amikacin, gentamicin, tobramycin), or quinolones (norfloxacin, ciprofloxacin); ^7^ lipopolysaccharide (LPS) mutant strains of *P. aeruginosa*; ^8^ multidrug-resistant strains; * in biofilm formation conditions.

**Table 8 nanomaterials-10-02022-t008:** Entirely polyester-based amino acid-modified cationic Ds developed recently and antibacterial activity of the most effective ones among the synthetized series.

Dendrimer Structure	Proposed Mechanism	HC_50_ (μg/mL)	Cytotoxicity IC_50_ (μg/mL)	Target Bacteria	Activity (μg/mL)
G1-G5 polyester-based alanine Ds NH_3_ or OH terminated [18]	OM/CM damage OM/CM disruption Leakage of cell contents	N.D.	G1/G2-NH_3_ NT 203 ^1,2,3^ G3-NH_3_ NT < 4.3 ^2,3^ <21.3 ^1^ G4-NH_3_ NT < 0.9 ^2,3^ <21 ^1^ G5-NH_3_ NT < 1.7 ^2,3^ <42.6 ^1^	*E. coli*	MIC G2-NH_3_ ^#^ 203.09
G1-G3 alanine and alkyl (C2-C14)-modified polyester-based dendrons (umbrella molecules) [96]	OM/CM-disrupting action Cationic surfactants-like activity Lytic mechanism	HC_50_ 10 G1/C14 63 G2/C14 5000 G3/C14	LC_50_ ^4^ 32 μg/mL G3/C14	*E. coli* *S. aureus* *P. aeruginosa* *A. baumannii* *MRSA* *E. faecalis*	MIC 3.9 (G1/C14) 3.9 (G2/C14) 7.8 (G3/C14) 3.9 (G1/C14) 1.95 (G2/C14) 3.9 (G3/C14) 5.5 (G3/C14) 11.0 (G3/C14) 5.5 (G3/C14) 8.0 (G3/C14)
5G polyester-based cationic Ds (G5K, G5H, G5HK) [1]	Electrostatic binding OM damage Non-lytic bactericidal mechanism	N.D.	Cells viability ^4^ 96.1%	*P. aeruginosa* (MDR) *P. aeruginosa* *P. aeruginosa ATCC* *P. putida* *P. florescens* *P. straminea* *A. baumannii* *A. pittii* *S. maltophilia*	MIC (μM) G5K ^#^ 2.1 2.1 2.1 1.0 0.5 1.0 1.1–2.1 2.1 1.1–4.2
				*P. florescens* *P. straminea* *A. baumannii 24* *A. pittii* *S. maltophilia 18*	MIC (μM) G5H ^#^ 8.3 8.3 8.3 8.3 8.3
				*P. aeruginosa 209* *P. aeruginosa 249* *P. aeruginosa ATCC* *P. putida* *P. florescens* *P. straminea* *A. baumannii* *A. pittii* *S. maltophilia 18* *S. maltophilia 11* *S. maltophilia 16* *S. maltophilia 19*	MIC (μM) G5HK ^#^ 4.2 4.2 8.4 2.1 1.0 1.0 4.2–8.4 2.1 2.1 4.2 8.4 8.4

N.D. = Not determined; K = *L*-lysine; H = *L*-histidine; N.T. = Not toxic; ^1^ RAW 264.7 cell lines; ^2^ HDF cell lines; ^3^ U87 cell lines; ^#^ MIC values are referred to the specific dendrimers named H1, bH1, G3KL and DG3KL in the original reference work because they were found to be the most active of a vast library of compounds; ^4^ HeLa cell line.

**Table 9 nanomaterials-10-02022-t009:** Most representative ammonium-terminated antibacterial Ds and some of their variations developed in the last decade and antibacterial activity of the most effective ones among the synthetized series.

Dendrimer Structure	Proposed Mechanism	HC_50_ (μg/mL)	Cytotoxicity IC_50_ (μg/mL)	Target Bacteria	Activity (μg/mL)
G2-G3 Ammonium-terminated carbosilane Ds [60]	Membrane impairments and disruption	N.D.	N.D.	*E. coli* *S. aureus* *E. coli* *S. aureus*	MIC 16 4 MBC 32 8
G2 Ammonium-terminated Ds [97]	Electrostatic binding CM permeabilization	1024 <10% HC	N.D.	*E. coli* *S. aureus*	MBC_99.9_ 3–8 4
G0-G2 Ammonium-terminated Ds [99]	Electrostatic binding Membrane damage Membrane disruption Leakage of cell contents	N.D.	N.D.	*E. coli* ATCC	MIC μg/mL; μM 0.9; 0.26 (G1) 1–16; 0.12–2 (G2) MBC μg/mL/μM 4–8; 1.1–2.3 (G1) N.D. (G2)
G1-G2 ammonium-terminated carbosilane dendrons **3**(G1), **4**(G2) G1-G2 ammonium-terminated carbosilane dendron nanocomposites **11**(3G1), **15**(3G2) G1-G2 ammonium-terminated carbosilane dendron nanocomposites **14**(4G1), **16**(4G2) [100]	Electrostatic binding Membrane damage Membrane disruption Leakage of cell contents	N.D.	N.D.	*E. coli*	MIC/MBC 16/16 (**3**) 4/8 (**4**) 16/16 (**11**) 16/16 (**15**) 16/32 (**14**) 4/4 (**16**)
				*S. aureus*	128/128 (**3**) 8/4-8 (**4**) 64/64 (**11**) 32/64 (**15**) 16/16-32 (**14**) 4-8/8 (**16**)
Ammonium-terminated Ru- and Zn encased Pcs Ds (RuPc, RuPc1, ZnPc, ZnPc1) [101]	Photoinactivating activity Oxidative stress induction by 1O_2_	N.D.	N.D.	*E. coli*	↓ 6-log10 (μM) ZnPc1 5 ^1^ ZnPc 2.5 ^1^
				*S. aureus*	ZnPc1 1 ^2^ ZnPc 1 ^2^

D(s) = dendrimer(s); N.D. = Not determined; ^1^ upon red light radiation of 30 J cm^−2^; ^2^ upon red light radiation of 60 J cm^−2^.

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
