# Peer review of "From Nanobiotechnology, Positively Charged Biomimetic Dendrimers as Novel Antibacterial Agents: A Review"

_nanomaterials, 2020, doi:10.3390/nano10102022_

Round 1
Reviewer 1 Report
Congratulations for your work. I think the work is very good, I will leave some poinst that could improve the manuscript:
- You do not need to demonstrate that your topic “cationic antibacterial dendrimers” has not enough publications. The editors and the reviewers will decided if the work brings a significant contribution to the field. If you want to demonstrated it will be more elucidated if it has only 2 collumns with the total of publiciations between “cationic antibacterial dendrimers” and “antimicrobial dendrimers”. This will be easier to read.
- I found some mistakes like “Fluorochinolones” the right way is fluoroquinolones.
- The quality of some figures in table 2 are low.
- In tables 3, 4 and so on the MIC should be always in the same unity and do not use sometimes micromolar other times μg/mL. I think they shoul always be micromolar but you can choose and use always the same unity.
- Tbale 7 and 8 are not clear. For example table 7 – what is the meaning of “MIC G3KL/DG3KL” in the last column. Or in the table 8 – “MIC (μM) G5H”. I understand that is not easier to explain in the table, but for the reader is even more difficult.
Author Response
Reviewer 1
Congratulations for your work. I think the work is very good, I will leave some points that could improve the manuscript:
1) You do not need to demonstrate that your topic “cationic antibacterial dendrimers” has not enough publications. The editors and the reviewers will decided if the work brings a significant contribution to the field. If you want to demonstrated it will be more elucidated if it has only 2 collumns with the total of publiciations between “cationic antibacterial dendrimers” and “antimicrobial dendrimers”. This will be easier to read.
The authors apologize to the Reviewer, but they cannot understand what the Reviewer's comment refers to. It probably concerns Figure 2, which compares, year by year, the number of publications on antimicrobial dendrimers (according to Scopus) with the more specific ones on cationic antibacterial dendrimers, topic of this review. Assuming that the authors, by inserting this Figure, did not want to prove anything about the meaning of their work, but simply intended to clarify to readers the state of the art in this still young research field, it is not clear what the Reviewer intends with "columns". The “columns” suggest a Table, but none of the Tables in the manuscript matches the Reviewer's comments. If by “columns”, he meant the bars of the histogram shown in Figure 2, the authors think that by reporting only the totality of publications on antimicrobial dendrimers and that on cationic antibacterial dendrimers, using a histogram with only two bars (“columns” as he says) as suggested by the reviewer, there would be a noticeable loss of information. The authors believe it is important to illustrate to readers the evolution of scientific interest over the years, also in terms of number of publications. Based on these considerations, the authors are confident that it is correct not to modify Figure 2 and to leave it in its original form.
2) I found some mistakes like “Fluorochinolones” the right way is fluoroquinolones.
The authors thank the reviewer for reporting the error. The erroneous name "fluoroquinolones" has now been changed to fluoroquinolones. Please see Table 1 (row six) and line 458.
3)The quality of some figures in table 2 are low.
The authors thank the Reviewer for his suggestion. Consequently, all the Figures in Table 2 have been improved in quality by working on size, on contrast, on brightness and optical resolution.
4) In tables 3, 4 and so on the MIC should be always in the same unity and do not use sometimes micromolar other times μg/mL. I think they shoul always be micromolar but you can choose and use always the same unity.
The reviewer's comment is correct and we very much agree, but unfortunately, many times it has not been possible to standardize the units of measurement of the MIC values, because the original reference paper did not report the molecular weight of dendrimers, necessary for the conversion of µg/mL in µM and vice versa. In this regard, the same authors have raised this important criticism, precisely in Section 5 (Conclusions and considerations of the authors). Please look at lines 1298-1305.
5) Tbale 7 and 8 are not clear. For example table 7 – what is the meaning of “MIC G3KL/DG3KL” in the last column. Or in the table 8 – “MIC (μM) G5H”. I understand that is not easier to explain in the table, but for the reader is even more difficult.
To meet the Reviewer's request, and to facilitate reading and understanding, footnotes have been inserted in Tables 7 and 8 to explain the meaning of the expressions used. See lines 762-763 (Table 7) and 1004-1005 (Table 8).

Reviewer 2 Report
The article titled “From nanobiotechnology, positively charged biomimetic dendrimers as novel antibacterial agents: a review” is a comprehensive attempt by the authors. However there are few comments to be addressed.
- Line No.55 – These molecules, without the need to enter the bacterium cell or to interfere with specific metabolic processes, basically act thanks to their positive charge
- Line No 58 - cytoplasmic membranes (OM and CM) and the killing of pathogens simply thanks to their contact with them, 59
- In introduction authors have written CAMPs, (Line 59, 60) - ….killing of pathogens simply thanks to their contact with them, thus bypassing the resistance mechanisms due to the different genetic mutations that bacteria can develop. This line need little more explanation.
- Line 64. 65 – “…the incapacity to permeate through the skin thanks to their macromolecular structure and high molecular weight (MW)” – Many places authors have repeatedly used the word ‘thanks’ meaninglessly in the manuscript.
- Line 152- Thanks to the presence of biofilm, bacteria growing inside it are 152 much more resistant to antimicrobial agents than planktonic forms and also their susceptibility to 153 antibiotics dramatically decreases. What authors want to convey here?
- Line 1051 – Rational for Ammonium termination to dendrimers is to be discussed.
- What is the reference for Figure 4, 5. ? Even though authors have discussed their views it is essential to cite the references.
Author Response
Reviewer 2
The article titled “From nanobiotechnology, positively charged biomimetic dendrimers as novel antibacterial agents: a review” is a comprehensive attempt by the authors. However there are few comments to be addressed.
1) Line No.55 – These molecules, without the need to enter the bacterium cell or to interfere with specific metabolic processes, basically act thanks to their positive charge
2)Line No 58 - cytoplasmic membranes (OM and CM) and the killing of pathogens simply thanks to their contact with them, 59
Since points 1 and 2 raised by the Reviewer simply quote sentences reported in the manuscript, the authors are confident that the actual request was included in point 3. As a result, the authors responded and satisfied the request contained in point 3. See point 3.
3) In introduction authors have written CAMPs, (Line 59, 60) - ….killing of pathogens simply thanks to their contact with them, thus bypassing the resistance mechanisms due to the different genetic mutations that bacteria can develop. This line need little more explanation.
To meet the Reviewer's request, the part of the main text, that the Reviewer reported as unclear has been rewritten. Furthermore, an additional sentence has been inserted to better explain the concepts expressed. Please look at the lines 56-65.
4) Line 64. 65 – “…the incapacity to permeate through the skin thanks to their macromolecular structure and high molecular weight (MW)” – Many places authors have repeatedly used the word ‘thanks’ meaninglessly in the manuscript.
The authors agree with the reviewer's comment. The repeated expression "thanks to" has been either removed (line 60) or replaced by the more adequate expression "due to" throughout the manuscript. Please see lines 69, 158, 423, 675, 879, 966, 1090, 1198, 1259, 1268, 1271.
5) Line 152- Thanks to the presence of biofilm, bacteria growing inside it are 152 much more resistant to antimicrobial agents than planktonic forms and also their susceptibility to 153 antibiotics dramatically decreases. What authors want to convey here?
With the quoted sentence, the authors wanted to explain that the bacteria in biofilm conditions (sessile forms) are more resistant to antimicrobial agents than bacteria that grow in normal conditions (planktonic forms), because the biofilm protect them, by several ways. For example, one way consists in making/ is to make it difficult to antibiotics to reach the bacteria that growth in the biofilm, by mechanically hampering their penetration through EPS, which is the main constituent of the biofilm. Table 1 reports in detail details other ways biofilms resist antimicrobials. However, the sentence has been rewritten to make it clearer and to satisfy the Reviewer. Please see lines 158-161.
6) Line 1051 – Rational for Ammonium termination to dendrimers is to be discussed.
The rational reason for Ammonium termination in dendrimers has been discussed, as requested. Please see lines 1067-1070 of the revised manuscript.
7) What is the reference for Figure 4, 5. ? Even though authors have discussed their views it is essential to cite the references .
References for Figure 4 and 5 have been inserted as requested. Please see lines 1225 and 1233.

Reviewer 3 Report
The review titled, "From nanobiotechnology, positively charged biomimetic dendrimers as novel antibacterial agents: a review" by Alfei and Schito is a more detailed and well studied one. In particular, the details of various dendrimers in tabulated form at appropriate places enhance readership feasibility and also indicates the authors detailed knowledge in the field. This review also discusses about the pros and cons of how various dendrimers were developed. The authors have also slightly debated about the usage of MIC and MBC concentration and their notations. Because the determination and representation of MIC values are very critical in the development of any antibacterial agent. Overall, I enjoyed reading the review and it would be definitely attract readers who are heading towards the antimicrobial dendrimers.Author Response
Reviewer 3
The review titled, "From nanobiotechnology, positively charged biomimetic dendrimers as novel antibacterial agents: a review" by Alfei and Schito is a more detailed and well studied one. In particular, the details of various dendrimers in tabulated form at appropriate places enhance readership feasibility and also indicates the authors detailed knowledge in the field. This review also discusses about the pros and cons of how various dendrimers were developed. The authors have also slightly debated about the usage of MIC and MBC concentration and their notations. Because the determination and representation of MIC values are very critical in the development of any antibacterial agent. Overall, I enjoyed reading the review and it would be definitely attract readers who are heading towards the antimicrobial dendrimers.
The authors warmly thank Reviewer 3 for not expressing substantial criticism on their work and for having judged it very positively.
